# COME: Adding Scene-Centric Forecasting Control to Occupancy World Model

**Yining Shi**[1, 2*], **Kun Jiang**[1, 2†], **Qiang Meng**[3], **Ke Wang**[3],
**Jiabao Wang**[4], **Wenchao Sun**[1, 2], **Tuopu Wen**[1, 2], **Mengmeng Yang**[1, 2†], **Diange Yang**[1, 2†]

[1]School of Vehicle and Mobility, Tsinghua University
[2]State Key Laboratory of Intelligent Green Vehicle and Mobility
[3]Kargobot Inc  [4]Nankai University

## Abstract

World models are critical for autonomous driving to simulate environmental dynamics and generate synthetic data. Existing methods struggle to disentangle ego-vehicle motion (perspective shifts) from scene evolvement (agent interactions), leading to suboptimal predictions. Instead, we propose to separate environmental changes from ego-motion by leveraging the scene-centric coordinate systems. In this paper, we introduce COME: a framework that integrates scene-centric forecasting Control into the Occupancy world ModEl. Specifically, COME first generates ego-irrelevant, spatially consistent future features through a scene-centric prediction branch, which are then converted into scene condition using a tailored ControlNet. These condition features are subsequently injected into the occupancy world model, enabling more accurate and controllable future occupancy predictions. Experimental results on the nuScenes-Occ3D dataset show that COME achieves consistent and significant improvements over state-of-the-art (SOTA) methods across diverse configurations, including different input sources (ground-truth, camera-based, fusion-based occupancy) and prediction horizons (3s and 8s). For example, under the same settings, COME achieves 26.3% better mIoU metric than DOME [4] and 23.7% better mIoU metric than UniScene [11]. These results highlight the efficacy of disentangled representation learning in enhancing spatio-temporal prediction fidelity for world models. Code is available at `https://github.com/synsin0/COME`.

## 1 Introduction

World models are designed to discern the current state of the environment and predict subsequent states based on executed actions. This predictive ability of world models not only enables the assessment of decision-making consequences but also facilitates the generation of synthetic data, which serves as a crucial resource for training, testing, and simulation in autonomous systems. Consequently, world models have emerged as a focal point of research in the field of autonomous driving, garnering substantial attention from both academia and industry.

The synthetic data generated by world models can take various forms to represent future scenes. These include 2D videos [6, 11, 24, 25], 3D lidar point clouds [1, 9, 11, 31, 37], and 3D occupancy grids [35, 8, 4, 33, 27, 11, 29, 26]. Irrespective of the representation format, the changes in the appearance of future scenes predicted by world models are predominantly governed by two key factors: 1) Ego-vehicle motion: The movement of the autonomous vehicle alters its viewing perspective,

---

* Work done during the internship at Kargobot Inc.
† Corresponding authors: Diange Yang, Kun Jiang, Mengmeng Yang

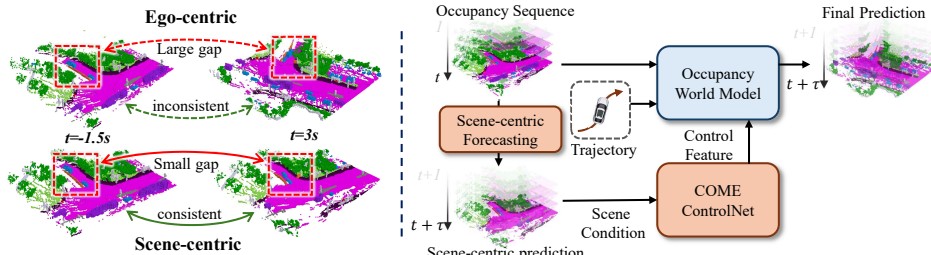

Figure 1: COME with both scene-centric and ego-centric representation. Compared to ego-centric evolution, scene-centric prediction shows smaller gap in the context of temporal evolution. COME uses scene-centric prediction results as an important guidance to enhance occupancy world model.

leading to dynamic changes in perceived spatial features (e.g., perspective shifts, occlusions). 2) Scene evolvement: Natural changes in the environment, such as agent interactions (e.g., pedestrian movements, vehicle collisions) and background updates (e.g., traffic light changes, weather variations). Notably, these two sources of change exhibit distinct characteristics. In a wide range of real-world scenarios, the background environment may remain relatively stable, with minimal changes over time (see Fig. 1). Instead, it is the motion of the ego-vehicle and the resulting shifts in the viewing perspective that contribute significantly to the dynamic changes in the world model's representation.

State-of-the-art (SOTA) world models, however, rely on neural networks to implicitly learn the intertwined effects of these factors, often resulting in suboptimal spatial consistency. As a case study, the SOTA model DOME [4] achieves an average 27.10 mIoU for 3-second occupancy prediction using ground-truth trajectories. In contrast, when predictions are formulated in a scene-centric coordinate system (where static backgrounds are decoupled from ego motion), a vanilla U-Net achieves 39.12 mIoU—a 44% improvement. This disparity underscores the critical lack of explicit control over spatial consistency in existing methods, motivating the need for disentangled representation learning.

To address this, we propose COME, a three-stage framework that leverages scene-centric coordinates to separate ego motion from scene dynamics (see Fig. 2): 1) Pose-conditioned generative diffusion stage: A diffusion-based model with spatiotemporal Diffusion Transformer generates future occupancy maps by iteratively denoising future occupancy latents, using historical occupancy latents and pose or BEV layouts as conditional inputs. 2) Fixed-view forecasting stage: By transforming past and future frames into a common coordinate system, this stage mitigates ego-motion effects, enabling non-interactive occupancy prediction without explicit scene flow estimation. 3) Forecasting guided diffusion stage: The COME ControlNet design acts as the core mechanism, transferring knowledge from fixed-view forecasts to variable-view generation. Structured as a trainable copy of the generative model's first half, the ControlNet injects scene-centric condition features into the latter half via skip connections, enhancing generative realism and temporal consistency.

We test under multiple configurations on nuScenes dataset and Waymo open dataset, and COME outperforms SOTA baselines by significant margins, with scene-centric settings demonstrating the largest gains. Key contributions include: 1) A divide-and-conquer strategy for occupancy world models, decomposing complex spatio-temporal prediction into ego-motion-agnostic and scene-dynamic components; 2) COME ControlNet, which leverages scene-centric forecasts as guidance to improve generative fidelity and spatial consistency; 3) Empirical validation of disentangled representation learning, establishing new state-of-the-art performance on a challenging autonomous driving benchmark.

## 2 Related Works

### 2.1 World Model in Autonomous Driving

Visual world models [6, 24, 25] leverage 2D video representations, offering great scalability due to the easy accessibility of camera data. However, the lack of 3D geometry understanding limits their fidelity in autonomous driving applications. In contrast, LiDAR-based representation [1, 9, 11, 31, 37] provides rich geometric comprehension but falls short in semantic-aware generation. Recently, occupancy representation [35, 8, 4, 33, 27, 11, 29, 26] have emerged as a compelling alternative for world modeling, due to their ability to encode both geometric and semantic information

simultaneously. Occupancy as intermediate representation also serves as strong geometric prior condition for downstream generation of driving videos[28] and LiDAR point clouds[11].

Some occupancy-based approaches [30, 33] leverage occupancy flow to forecast future scenarios. While achieving promising results, they often depend on additional annotations and struggle to produce imaginative predictions. Recent works have shifted toward generative frameworks [35, 4, 27, 26]: for example, OccWorld [35] employs a two-stage pipeline, first tokenizing occupancy with a VQ-VAE [23] and then predicting ego motion and scene evolution autoregressively. Inspired by the success of large language models, some works [27, 26] further enhance the model interpretability during generation. Recent methods [4, 11] employ diffusion transformers (DiTs), demonstrating strong generative capabilities for world modeling. In this work, we similarly adopt the DiT paradigm while introducing control features inspired by ControlNet [34] — a pioneering framework for conditional generation in 2D images using multi-modal inputs (e.g., depth maps, edges, or sketches). By incorporating explicit control signals, our method achieves superior spatial consistency and prediction accuracy, establishing new state-of-the-art performance for occupancy world models.

## 2.2 4D Occupancy Forecasting

Occupancy world models and 4D occupancy forecasting methods share a common paradigm of predicting future occupancy states. The key distinction is that occupancy forecasting typically targets short-term scene evolution and does not involve predicting future ego trajectories. Occ4Cast [15] and Cam4DOcc [17] establish LiDAR-based and camera-based benchmarks, respectively, and propose baseline models using temporal recurrent networks such as ConvLSTM [19]. To address the high cost of 4D occupancy annotation, 4D-Occ-Forecasting [10] leverages future point clouds as proxies for future occupancy and employs differentiable depth rendering for self-supervised learning. Building upon this, Vidar [32] and UnO [2] introduce latent rendering and continuous 4D fields, respectively, to further enhance self-supervised learning for 4D occupancy prediction and related downstream tasks. In our work, the scene-centric prediction module adopts a similar paradigm to 4D occupancy forecasting - predicting future occupancy while explicitly factoring out the influence of ego trajectory. This design establishes a spatially consistent control prior that effectively guides the learning process.

## 3 Methodology

This section presents our COME, a framework that integrates scene-centric forecasting Control into the Occupancy world ModEl. We detail the three main components of COME (illustrated in Fig. 2) in Secs. 3.1 to 3.3. Finally, Sec. 3.4 describes the training objectives and the overall training pipeline.

## 3.1 Occupancy World Model

We first describe our modified baseline world model, which is capable of performing the generation task independently. Following DOME [4], our model leverages diffusion Transformers (DiTs) for the superior fine-grained and imaginative generation compared to auto-regressive counterparts [35, 4, 27, 26]. Given historical observations $\mathbf{x}_{1:t}$, ego-vehicle states $p_{1:t}$ and other possible inputs (e.g., BEV layouts), the occupancy world model predicts future occupancy $\hat{\mathbf{x}}_{t+1:t+\tau}$ while optionally forecasting future trajectories $\hat{p}_{t+1:t+\tau}$. We treat future trajectories — whether ground-truth or predicted by a planning module — as available inputs for the generative model for two key reasons: (1) It ensures comparability with existing methods, since some directly generate with ground-truth trajectories while others predict by themselves. (2) It enables trajectory-conditioned generation, which is essential for world models to produce diverse and controllable outputs.

Our model architecture comprises two principal components. The first is a collection of input encoders that transform historical observations into compact representations for efficient processing during the diffusion stage. We adopt the trajectory encoder from DOME [4] to encode trajectories, and apply a max-pool layer for BEV layouts when available. For the occupancy data, we utilize a Occupancy Variational Auto-Encoder (Occ-VAE) consisting of an encoder network $q_\phi(\mathbf{z}|\mathbf{x})$ and a decoder network $p_\theta(\mathbf{x}|\mathbf{z})$. Given input occupancy $\mathbf{x} \in \mathcal{R}^{H \times W \times D}$, $q_\phi(\mathbf{z}|\mathbf{x})$ encodes it into a continuous latent variable $\mathbf{z} \sim q_\phi(\mathbf{z}|\mathbf{x})$. Conversely, $p_\theta(\mathbf{x}|\mathbf{z})$ is able to reconstruct the occupancy from the latent variable. This design enables the Occ-VAE to compress high-dimensional occupancy data into a compact latent space, facilitating more efficient processing in the generative model.

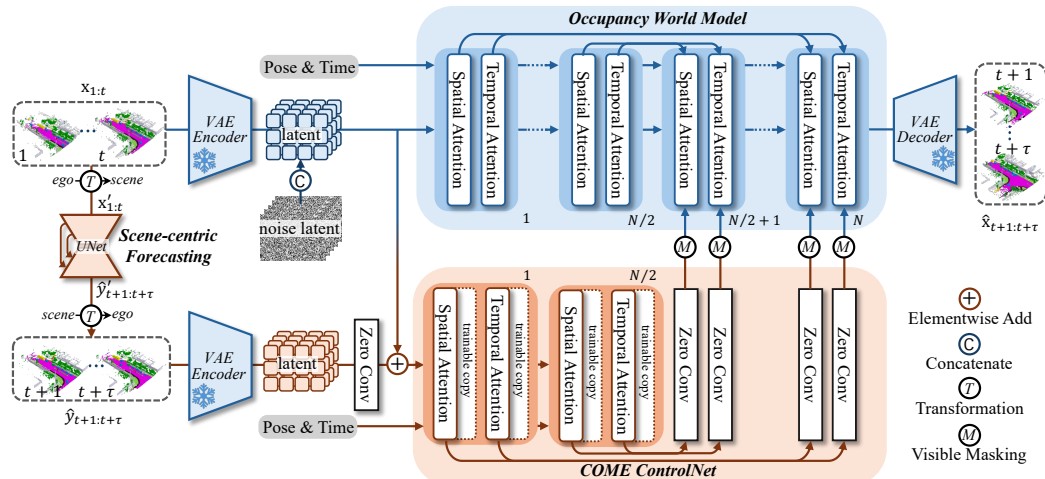

Figure 2: The proposed COME framework comprises three main modules: (1) an **Occupancy World Model** that predicts future occupancy using historical observations and other inputs (e.g., poses, time steps, BEV layouts); (2) a **Scene-centric Forecasting Module** that produces spatially consistent scene predictions by eliminating the influence of ego motion; and (3) the **COME ControlNet** which converts the scene conditions from the forecasting module into control features that are subsequently injected into the world model for controllable and geometrically coherent occupancy generation.

The second component is the spatial-temporal Diffusion Transformer modified from the DOME [4], which consists of alternating stacked spatial and temporal layers. To reserve interface for later feature conditioning, we introduce skip connections between every early and late layer pair, similar to the architectural designs in UNet [18] and HunyuanDiT [13]. For example, the last spatial block takes the first spatial block as a skip input. For each block in the second half, the input and the skip are concatenated in channel dimension and undergo a simple MLP for feature fusion.

The model processes latent features encoded from occupancy data, along with optional inputs, e.g. BEV layouts if available. Trajectories are converted into trajectory condition and injected into each spatial-temporal block, enabling trajectory-aware generation. In the end, the outputs are decoded through $q_\phi(\mathbf{z}|\mathbf{x})$ into occupancy predictions conditional to various future time steps and ego poses.

## 3.2 Scene-centric Forecasting Module

Future occupancy prediction represents a complex spatial-temporal modeling challenge that requires reasoning about two key factors: (1) the natural evolution of the scene (e.g., moving objects, infrastructure changes), and (2) the planned trajectory of the ego-vehicle. Our foundation world model described in Sec. 3.1 treats the future trajectory as a condition but has limited understanding of the inherent nature of scene evolution, leading to spatially inconsistent scene evolution in the generated occupancy — a limitation also confirmed in later experiments. To address this, we propose a scene-centric forecasting module that produces more coherent scene evolution and generates scene-conditioned inputs for the world model.

In practice, the scene exhibits relatively smaller changes when represented in a unified coordinate frame compared to an ego-centric occupancy representation. In Occ3D-nuScenes, static voxels account for over 92.7% of all occupied voxels, suggesting that forecasting the scene in a shared coordinate system simplifies the task without requiring complex designs. In our implementation, we find that a simple UNet [18] suffices for high-quality future predictions, where the skip connections effectively preserve static voxels, while the multi-scale structure captures dynamic elements.

To generate module inputs, we transform the historical occupancy sequence $\mathbf{x}_{1:t}$ into a unified coordinate frame at timestamp $t$. Each frame $\mathbf{x}_i$ is transformed using ego-vehicle states: $\mathbf{x}'_i = T(\mathbf{x}_i, p_i, p_t)$ for $i = 1, 2, \cdots, t$. This transformation is implemented by initializing an empty grid $\mathbf{x}'_i \in \mathbb{R}^{D \times H \times W}$ and populating each voxel via nearest-neighbor sampling from $\mathbf{x}_i$, using

the coordinate transformation from $p_t$ to $p_i$. The resulting aligned sequence $\mathbf{x}'_{1:t}$ is stacked into a $Dt \times H \times W$ tensor, which serves as the module input.

Subsequently, our UNet produces an output tensor of dimensions $D\tau \times H \times W$, which is decoded to obtain the future occupancy sequence $\hat{\mathbf{y}}'_{t+1:t+\tau}$. These predictions, formulated in the coordinate frame of ego-pose $p_t$, are then transformed to the vehicle's future poses via $\hat{\mathbf{y}}_i = T(\hat{\mathbf{y}}'_i, p_t, p_i)$. Each resulting occupancy $\hat{\mathbf{y}}_i$ is encoded by the Occ-VAE encoder, producing the final scene conditions $\mathbf{z}_i = q_\phi(\mathbf{z}|\hat{\mathbf{y}}_i)$, for $i = t+1, t+2, \cdots, t+\tau$.

### 3.3 COME ControlNet

The COME ControlNet, which encodes scene conditions $\{\mathbf{z}_i\}_{i=1}^\tau \in \mathbb{R}^{C \times H_1 \times W_1}$ into implicit control features, consists of $N$ blocks inspired by HunyuanDiT[13]. As shown in Fig. 2, the first $N/2$ blocks are trainable copies of the corresponding spatial-temporal blocks from the occupancy world model, while the remaining blocks are the zero convolution layers. Each of these layers outputs control feature $\{\mathbf{c}_i^n\}_{i=1}^\tau \in \mathbb{R}^{C \times H_1 \times W_1}$ for $n = N/2 + 1, \cdots, N$, where $\tau$ denotes the number of future frames, and $C, H_1, W_1$ represent the channel depth, height, and width dimensions, respectively.

However, we observe that $\{\mathbf{c}_i^n\}_{i=1}^\tau$ is not directly suitable as a control prior for world model. This limitation arises because the forecasting module, due to its simple structure, lacks sufficient capacity for imagining future states. Consequently, its predictions in historically unobserved regions may bring noise, degrading the world model's generative performance — a finding corroborated by late experiments. Thus, filtering unreliable features from control features is crucial for robust generation.

To address this, we propose a visibility-aware masking strategy based on 3D spatial relationships. For each future timestamp $i \in \{t+1, t+2, \cdots, t+\tau\}$, we first trace the root source of the control feature $\mathbf{c}_i^n$, namely the $\hat{\mathbf{y}}_i$ predicted by the scene-centric forecasting module. We then construct a binary voxel mask $\mathbf{m}_i \in \mathbb{R}^{D \times H \times W}$, where a value of 1 indicates that the corresponding voxel center in $\hat{\mathbf{y}}_i$ is observable in historical occupancy data, and 0 otherwise. This allows us to derive an invisibility mask $\mathbf{M}_i \in \mathbb{R}^{H_1 \times W_1}$ that quantifies the reliability of each spatial feature in $\mathbf{c}_i^n$:

$$\mathbf{M}_i(h, w) = \mathbb{I}\left(\frac{1}{D \cdot \delta_h \cdot \delta_w} \sum_{d=1}^{D} \sum_{u=h \cdot \delta_h}^{(h+1) \cdot \delta_h} \sum_{v=w \cdot \delta_w}^{(w+1) \cdot \delta_w} \mathbf{m_i}(d, u, v) < \varepsilon\right), \qquad (1)$$

where $\delta_h = H/H_1, \delta_w = W/W_1$ and $\varepsilon$ is a pre-defined threshold. In practice we set $\varepsilon = 0.5$. The $\mathbb{I}$ is an indicator function to check whether proportion of historically observed voxels within each pillar (corresponding to feature location (h, w) in $\mathbf{c}_i^n$) is large enough. Then, we suppress unreliable features via element-wise multiplication: $\mathbf{c}_i^n \leftarrow \mathbf{c}_i^n \odot \mathbf{M}_i$. Finally, these refined control features are injected into the world model through residual addition at corresponding layers, ensuring robust and controllable generation. The control features are added to skip features and concatenated together to the input features for each blocks in the second half of the world model.

### 3.4 Training Pipelines

Inspired by ControlNet [34], we employ a multi-stage training pipeline as (1) In **stage 1**, the occupancy world model is trained with configurations introduced in DOME [4]. This stage establishes a strong foundational occupancy generation ability. (2) In **stage 2**, we train the UNet-based forecasting module using a simple cross-entropy loss. (3) In **stage 3**, we freeze all other modules and exclusively train the parameters of ControlNet. Our multi-stage training strategy not only optimizes training efficiency but also ensures controllable generation, as demonstrated in our ablation studies.

## 4 Experiments

We elaborate our experimental settings in Sec. 4.1, the quantitative and qualitative results of the proposed COME framework in Secs. 4.2 and 4.3, and extensive ablation studies in Sec. 4.4.

### 4.1 Experimental Setup

**Dataset and metrics** Most experiments are conducted on the widely used Occ3D-nuScenes[22] benchmark, which offers 3D occupancy labels for 18 categories based on the large-scale nuScenes[3]

Table 1: **4D occupancy generation performance under various settings.** Each setting varies in terms of input modality and whether the ego trajectory (ego traj.) is predicted (Pred.) by a planning module or provided as ground truth (GT). "Avg." indicates the average performance across 1s, 2s, and 3s horizons. The best performance in each setting is highlighted in bold.

| Method | Input | Ego traj. | mIoU (%) ↑ | | | | IoU (%) ↑ | | | |
|---|---|---|---|---|---|---|---|---|---|---|
| | | | 1s | 2s | 3s | Avg. | 1s | 2s | 3s | Avg. |
| OccWorld-D [35] | Camera | Pred. | 11.55 | 8.10 | 6.22 | 8.62 | 18.90 | 16.26 | 14.43 | 16.53 |
| OccWorld-T [35] | Camera | Pred. | 4.68 | 3.36 | 2.63 | 3.56 | 9.32 | 8.23 | 7.47 | 8.34 |
| OccWorld-S [35] | Camera | Pred. | 0.28 | 0.26 | 0.24 | 0.26 | 5.05 | 5.01 | 4.95 | 5.00 |
| OccWorld-F [35] | Camera | Pred. | 8.03 | 6.91 | 3.54 | 6.16 | 23.62 | 18.13 | 15.22 | 18.99 |
| OccLLaMA [26] | Camera | Pred. | 10.34 | 8.66 | 6.98 | 8.66 | 25.81 | 23.19 | 19.97 | 22.99 |
| OccVAR [8] | Camera | Pred. | 17.17 | 10.38 | 7.82 | 11.79 | 27.60 | 25.14 | 20.33 | 24.35 |
| DFIT-OccWorld [33] | Camera | Pred. | 13.38 | 10.16 | 7.96 | 10.50 | 19.18 | 16.85 | 15.02 | 17.02 |
| Occ-LLM-S [27] | Camera | Pred. | 11.28 | 10.21 | 9.13 | 10.21 | 27.11 | 24.07 | 20.19 | 23.79 |
| RenderWorld-S [29] | Camera | Pred. | 2.83 | 2.55 | 2.37 | 2.58 | 14.61 | 13.61 | 12.98 | 13.73 |
| COME (Ours) | Camera | Pred. | **25.57** | **18.35** | **13.41** | **19.11** | **45.36** | **37.06** | **30.46** | **37.63** |
| DOME [4] | Camera | GT | 24.12 | 17.41 | 13.24 | 18.25 | 35.18 | 27.90 | 23.44 | 28.84 |
| COME (Ours) | Camera | GT | **26.56** | **21.73** | **18.49** | **22.26** | **48.08** | **43.84** | **40.28** | **44.07** |
| Copy&Paste [35] | 3D-Occ | Pred. | 14.91 | 10.54 | 8.52 | 11.33 | 24.47 | 19.77 | 17.31 | 20.52 |
| OccWorld [35] | 3D-Occ | Pred. | 25.78 | 15.14 | 10.51 | 17.14 | 34.63 | 25.07 | 20.18 | 26.63 |
| OccLLaMA [26] | 3D-Occ | Pred. | 25.05 | 19.49 | 15.26 | 19.93 | 34.56 | 28.53 | 24.41 | 29.17 |
| OccVAR [8] | 3D-Occ | Pred. | 27.96 | **21.75** | 16.47 | 22.06 | 38.73 | 29.50 | 24.86 | 31.03 |
| RenderWorld [29] | 3D-Occ | Pred. | 28.69 | 18.89 | 14.83 | 20.80 | 37.74 | 28.41 | 24.08 | 30.08 |
| Occ-LLM [27] | 3D-Occ | Pred. | 24.02 | 21.65 | **17.29** | 20.99 | 36.65 | **32.14** | **28.77** | **32.52** |
| DFIT-OccWorld [33] | 3D-Occ | Pred. | **31.68** | 21.29 | 15.18 | **22.71** | **40.28** | 31.24 | 25.29 | 32.27 |
| COME (Ours) | 3D-Occ | Pred. | 30.57 | 19.91 | 13.38 | 21.29 | 36.96 | 28.26 | 21.86 | 29.03 |
| DOME [4] | 3D-Occ | GT | 35.11 | 25.89 | 20.29 | 27.10 | 43.99 | 35.36 | 29.74 | 36.36 |
| COME (Ours) | 3D-Occ | GT | **42.75** | **32.97** | **26.98** | **34.23** | **50.57** | **43.47** | **38.36** | **44.13** |
| UniScene-Fore [11] | 3D-Occ(2f),Box,Map | GT | 35.37 | 29.59 | 25.08 | 31.76 | 38.34 | 32.70 | 29.09 | 34.84 |
| COME (Ours) | 3D-Occ(2f),Box,Map | GT | **45.98** | **38.57** | **33.28** | **39.28** | **52.11** | **46.73** | **42.65** | **47.16** |

dataset. The annotations cover a spatial range of [-40m, -40m, -3.2m, 40m, 40m, 3.2m] with a voxel resolution of [0.4m, 0.4m, 0.4m], resulting in a $200 \times 200 \times 16$ voxel grid per frame. The dataset is split into 700 training, 150 validation, and 150 test driving sequences, each lasting 20 seconds.

We also use Occ3D-Waymo[22] benchmark based on the Waymo Open Dataset[21] (WOD), which has 3D occupancy labels for 16 categories. The spatial range and voxel resolution are the same as on nuScenes dataset. The dataset is split into 798 training and 202 validation driving sequences. To align with the task settings on nuScenes, we down-sample to select one frame every 0.5 seconds.

We adopt geometric Intersection over Union (IoU) and semantic mean Intersection over Union (mIoU) as evaluation metrics. Results are reported at each future timestamp, along with the average performance over all timestamps on the validation set, in line with previous works.

**Implementation details.** Unless otherwise specified, our world model predicts future occupancy over a 3-second horizon at 2 frames per second, conditioned on 4 frames of historical occupancy data, following the protocol established in OccWorld [35]. As outlined earlier, the COME framework is trained in multiple stages: (1) *Diffusion-based World Model.* We adopt the pre-trained Occ-VAE from DOME [4] and train the diffusion-based world model for 2000 epochs with a total batch size of 128 and a learning rate of 2e-4. (2) *Scene-centric Forecasting Module.* This module is trained for 12 epochs using a total batch size of 32 and the CBGS resampling strategy [36]. (3) *COME ControlNet.* This component is trained for 1000 epochs with a total batch size of 64. All models are trained on 4 H20 GPUs and use a learning rate of 1e-4 is not stated specifically. Please refer to the supplementary materials for additional implementation details, including network architectures and statistics, training hyperparameters, and planning trajectory configurations.

### 4.2 Quantitative Evaluation

**Main results on Occ3D-nuScenes.** In the occupancy world modeling domain, existing methods differ significantly in their experimental setups. To enable a fair and comprehensive comparison, we

Table 2: Comparisons of 4D occupancy forecasting performance between DOME and different stages of COME on Occ3D-Waymo dataset.

| Method | mIoU (%) ↑ | | | | IoU (%) ↑ | | | |
|---|---|---|---|---|---|---|---|---|
| | 1s | 2s | 3s | Avg | 1s | 2s | 3s | Avg |
| DOME | 31.62 | 24.82 | 19.65 | 25.36 | 51.24 | 43.17 | 37.47 | 43.96 |
| COME Stage 1: World Model | 34.45 | 26.28 | 22.50 | 27.74 | 51.16 | 43.24 | 38.44 | 44.28 |
| COME Stage 2: Scene-Centric Forecasting | **47.37** | **40.85** | 36.31 | **41.51** | 59.69 | 52.27 | 45.60 | 52.52 |
| COME Stage 3: ControlNet | 39.85 | 34.65 | 30.58 | 35.03 | 55.53 | 50.09 | 45.53 | 50.38 |
| COME+ Stage 3: ControlNet | 47.29 | 40.80 | **36.44** | **41.51** | **60.63** | **53.98** | **48.62** | **54.41** |

adapt our proposed COME framework to match these various settings and report the main results in Tab. 1.

The first setting uses camera images as input and predicted ego trajectories from a planning module. In this case, we employ a modified BEVDet [7] to convert camera inputs into occupancy predictions. Among prior works, OccVAR previously achieved the best performance with an average mIoU of 11.79 and an average IoU of 24.35. In contrast, our COME achieve 19.11 mIoU and 37.63 IoU, significantly surpassing the previous best results by 62.1% and 54.5%, respectively.

When ground-truth ego trajectories are used with the same camera input, our method further improves performance to 22.26 mIoU and 44.07 IoU, with gains of 3.15 and 6.44, respectively. In this setting, our method also outperforms the state-of-the-art DOME by a significant margin 22.0% in mIoU and 34.6% in IoU, highlighting the robustness of COME.

In the third setting, we adopt ground-truth occupancy inputs but use predicted ego trajectories. Here, the best prior mIoU (22.71) is achieved by DFIT-OccWorld, while the highest IoU (32.52) is obtained by Occ-LLM. It is noteworthy that both methods incorporate additional cues such as occupancy flow or language information into their frameworks. In comparison, our COME, despite relying solely on trajectory conditioning without such external information, still achieves competitive results of 21.29 mIoU and 29.03 IoU. We hypothesize that the slightly lower performance is due to COME's strong dependency on trajectory input: since it generates occupancy strictly conditioned on predicted trajectories, it may be more sensitive to planning errors. We further validate the hypothesis in the ablation study. Nevertheless, our method remains among the top-performing approaches, demonstrating its effectiveness even in this challenging setup.

Under the configuration with both ground-truth occupancy and ground-truth trajectories, COME achieves 34.23 mIoU and 44.13 IoU, showing strong performance when free from upstream prediction errors. COME outperforms the state-of-the-art DOME by 26.3% in mIoU and 21.3% in IoU. Furthermore, with BEV layouts, COME's performance increases further to 39.28 mIoU and 47.16 IoU. COME outperforms the state-of-the-art UniScene-Fore by 23.7% in mIoU and 35.4% in IoU.

**Results on Occ3D-Waymo.** On Occ3D-Waymo, we retrain OCC-VAE, UNet[18], DOME[4], COME world model, and COME ControlNet on Occ3d-Waymo, initializing from nuScenes-trained components to save cost and time, with unchanged model sizes and training schedules.

We report the default COME model and a COME variant (COME+) with the same structure and with stronger geometric controls from the scene-centric forecasting module, differing by additionally replacing noisy BEV latents with UNet-encoded latents in future visible areas during denoising.

The Occ-VAE reconstruction quality on Occ3D-Waymo is $mIoU = 74.88$ and $IoU = 82.57$.

The results of Occ3D-Waymo are shown in Table 2. our reproduced DOME achieves similar performance as in nuScenes dataset. COME world model has marginally better performance with DOME by 2.38 mIoU and 0.32 IoU. Scene-centric forecasting module outperforms the generative world model by a large margin (63% mIoU and 19.5% IoU). COME achieves 38.1% better mIoU and 14.6% better IoU compared to DOME[4]. COME+, which relies more heavily on condition features, maintains the same 3-s average mIoU with Scene-centric Forecasting module and further improves 3-s average IoU to 23.7% improvement.

Table 3: **Long-term 4D occupancy generation performance.** Ground-truth 3D occupancy and trajectories are used as inputs. The best results are highlighted in bold. "Avg." denotes the average performance over the 1 second to 8 second horizon.

| Method | mIoU (%) ↑ | | | | | | | | |
| | 1s | 2s | 3s | 4s | 5s | 6s | 7s | 8s | Avg. |
|---|---|---|---|---|---|---|---|---|---|
| DOME [4] | 30.10 | 21.35 | 17.36 | 14.86 | 12.61 | 11.03 | 10.00 | 9.34 | 15.83 |
| COME (Ours) | **33.78** | **24.57** | **21.35** | **18.25** | **15.84** | **13.85** | **12.99** | **11.96** | **19.07** |

| Method | IoU (%) ↑ | | | | | | | | |
| | 1s | 2s | 3s | 4s | 5s | 6s | 7s | 8s | Avg. |
|---|---|---|---|---|---|---|---|---|---|
| DOME [4] | 39.04 | 31.20 | 27.14 | 24.73 | 22.32 | 20.28 | 19.05 | 17.97 | 25.21 |
| COME (Ours) | **44.20** | **36.25** | **32.86** | **30.03** | **26.93** | **24.70** | **23.30** | **21.44** | **29.96** |

Table 4: **Model performance in various stages.** Here, "Visible", "Invisible" and "All" refer to voxels that are observed, unobserved, and the total set of voxels based on historical occupancy, respectively.

| Model | Input | mIoU (%) ↑ | | | IoU (%) ↑ | | |
| | | Visible | Invisible | All | Visible | Invisible | All |
|---|---|---|---|---|---|---|---|
| Stage1: World Model | 3D-Occ (4f) | 25.68 | 5.81 | 23.58 | 35.96 | 13.60 | 32.61 |
| Stage2: Scene-Centric Forecasting | 3D-Occ (4f) | 42.74 | 0.09 | 39.12 | 55.08 | 0.31 | 48.00 |
| Stage3: ControlNet | 3D-Occ (4f) | 40.06 | 5.56 | 34.23 | 51.12 | 14.95 | 44.13 |
| Stage1: World-Model | 3D-Occ(2f),Box,Map | 32.92 | 11.74 | 30.04 | 39.52 | 16.20 | 35.43 |
| Stage2: Scene-Centric Forecasting | 3D-Occ(2f) | 41.65 | 0.08 | 37.93 | 54.50 | 0.27 | 47.18 |
| Stage3: ControlNet | 3D-Occ(2f),Box,Map | 42.75 | 14.85 | 39.28 | 52.78 | 20.48 | 47.16 |

**Long-term Occupancy Generation** The ability to generate long-term predictions is crucial for world models, particularly in the context of autonomous driving. To evaluate this capability, we extend the prediction horizon from 3 seconds to 8 seconds and present the results in Tab. 3. Our method, COME, consistently outperforms baselines across all timestamps and on average, for both mIoU and IoU metrics. Specifically, COME achieves average mIoU and IoU scores of 19.07 and 29.96, surpassing DOME by 20.5% and 18.8% respectively.

We further discuss in the supplementary material that the best practice of masking strategy for balancing quantitative and qualitative results is to pose the invisibility mask on control features.

## 4.3 Qualitative Results

Fig. 3 presents visualizations comparing ground-truth occupancy with predictions from DOME and our method. We observe that DOME suffers from object category inconsistency (top example) and sudden object disappearance (bottom example). In contrast, COME produces results with improved spatial consistency, highlighting the efficacy of the scene condition injection. Additional qualitative results, including analyses of input variants, model architectures, and component contributions, are provided in the supplementary material for comprehensive evaluation.

## 4.4 Ablation Study

**Model performances across various stages.** In Tab. 4, we analyze the predicted occupancy across our three stages, evaluating visible, invisible, and all voxels. Consistently, the world model performs better in previously invisible areas, while scene-centric forecasting excels in observed regions. This validates our motivation: the generative world model exhibits strong imaginative capabilities but under-utilizes the 3D spatial consistency of the driving scene. In contrast, scene-centric forecasting achieves high accuracy in observed areas but lacks generative flexibility, limiting its imagination applicability to novel-view synthesis under diverse trajectories.

In the third stage, we leverage the ControlNet to combine the strengths of the two modules. Stage three model significantly improves performance in invisible regions compared to the forecasting module. Although the overall mIoU experiences a slight decline, the model gains imaginative capabilities and broader adaptability. With extra BEV layout inputs, COME demonstrates further

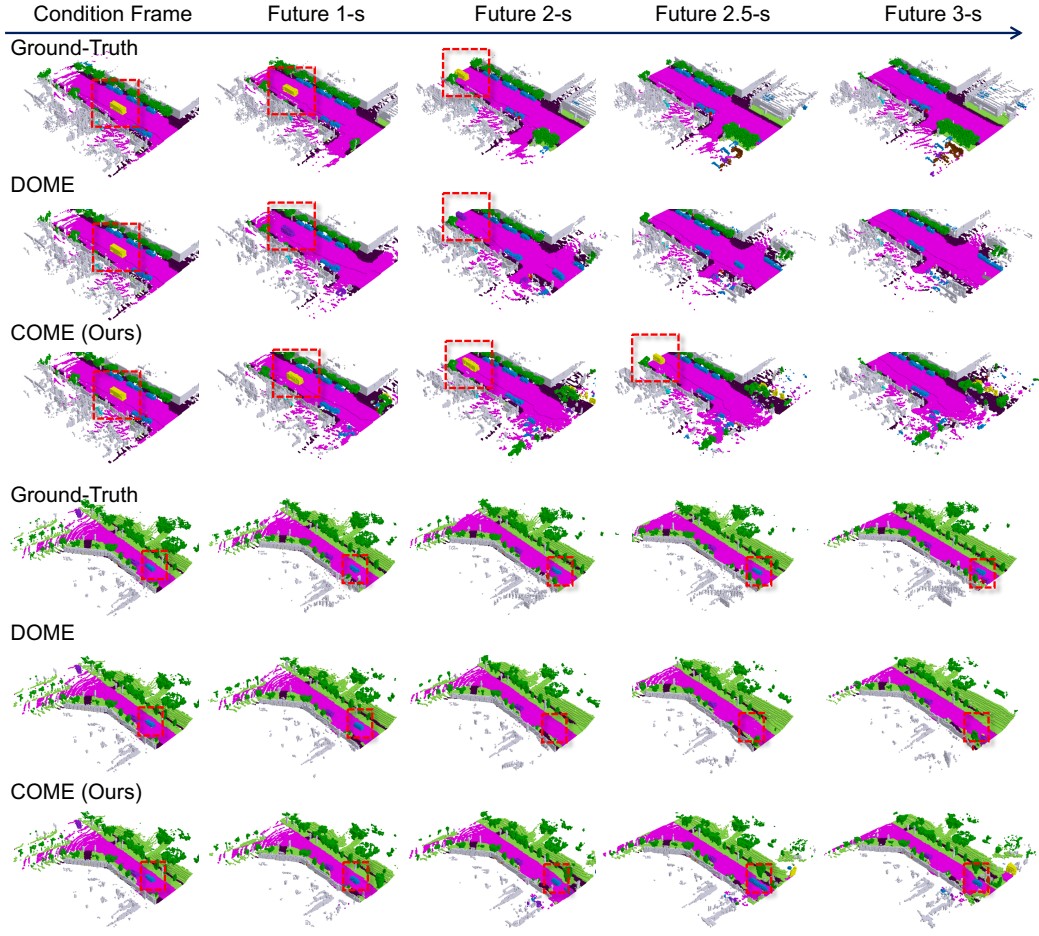

Figure 3: Qualitative results of 3-s 4D occupancy generation.

enhanced generation of invisible scenes and achieves the best overall performance. These results showcase the effectiveness of our proposed framework.

**Effects of training setups.** Tab. 5 presents an ablation study on different training configurations within the proposed COME framework. We first train ControlNet with and without freezing the world model in the final stage, and present the results in Tab. 5a. It can be observed that training ControlNet while keeping the world model frozen yields significantly better results, confirming the importance of aligning with the fine-tuning strategy validated in the original ControlNet [34] paper.

In Tab. 5b, we explore two model sizes for both the generative model and its corresponding Control-Net. Across both settings, the introduction of ControlNet consistently improves performance by a substantial margin. These gains are particularly pronounced under limited computational budgets, where mIoU increases from 7.78 to 32.00 and IoU from 18.14 to 42.03 — surpassing even the standalone world model with 1375.4 GFLOPS. These results demonstrate that integrating ControlNet significantly enhances the generative capability of the model even if the generation model is small.

**Effects of inference setups.** In Tab. 6, we investigate how different inference configurations affect model performance. We begin by analyzing the impact of the number of denoising steps during the diffusion process, as shown in Tab. 6a. Increasing the number of steps consistently improves generative performance. The results indicate a clear positive correlation between performance and the number of denoising steps, with satisfactory results achieved when performing at least 10 steps.

Next, we examine model performance using different sources of predicted occupancy and trajectories. In Tab. 6b, we replace ground-truth occupancy with predictions from a vision-only model, BEVStereo

Table 5: Ablations on different training setups. "WM" and "ControlNet" denote the world model and ControlNet in our COME, respectively.

(a) Effects of trainable parameters in the final stage.

| Trainable modules | mIoU | IoU |
|---|---|---|
| ControlNet | 34.23 | 44.13 |
| WM&ControlNet | 28.67 | 41.07 |

(b) Model performances under different model sizes.

| Model | Small | | | Base | | |
|---|---|---|---|---|---|---|
| | mIoU | IoU | GFLOPS | mIoU | IoU | GFLOPS |
| WM | 7.78 | 18.14 | 147.8 | 23.49 | 32.36 | 1375.4 |
| +ControlNet | 32.00 | 42.03 | 222.7 | 34.23 | 44.13 | 2066.6 |

Table 6: Ablations on various inference configurations.

(a) Effects of the denoising steps at the inference stage.

| #step | mIoU | IoU |
|---|---|---|
| 2 | 4.41 | 14.47 |
| 5 | 4.95 | 16.17 |
| 10 | 33.42 | 44.35 |
| 20 | 34.23 | 44.13 |

(b) Model performance when ground-truth occupancy is replaced with different occupancy generators. Here, "C" and "L" denote inputs from camera and LiDAR sensors, respectively.

| Model | Input | mIoU | IoU |
|---|---|---|---|
| BEVStereo [12] | C | 22.26 | 44.07 |
| EFFOcc [20] | LC | 26.75 | 50.49 |

(c) Effects of the used trajectories. "align." is the alignment operation of the ground-truth occupancies.

| Pose2D | Yaw | Align. | mIoU | IoU |
|---|---|---|---|---|
| GT | GT | - | 34.23 | 44.13 |
| Pred. | GT | - | 25.90 | 35.21 |
| Pred. | Pred. | - | 21.29 | 29.03 |
| Pred. | Pred. | ✓ | 34.00 | 43.69 |

(mIoU = 42.54), and a fusion model, EFF-Occ (mIoU = 54.08). The results show that stronger occupancy inputs lead to better generative outcomes. Tab. 6c shows that gradually replacing the ground-truth pose and yaw with predicted values leads to a noticeable drop in model performance. However, when we remove the influence of ego pose during evaluation - by aligning predicted and ground-truth occupancy to a same predicted future waypoint coordinate - the degradation is minimal, with only a 0.23 drop in mIoU and a 0.44 drop in IoU. This phenomenon suggests that the performance degradation primarily stems from misaligned trajectories, while the generated scene quality remains relatively unaffected by trajectory errors. Further discussion is provided in our supplementary material.

## 5  Conclusion

We introduce COME, a framework that enhances generative occupancy world models through scene-centric forecasting control. By explicitly decoupling ego-motion effects from scene evolution, COME first generates spatially consistent, ego-invariant control features, which are then integrated into the occupancy world model for more accurate and controllable future predictions. Extensive experiments on the large-scale Occ3D-nuScenes dataset demonstrate the state-of-the-art performances across multiple settings, validating the effectiveness of our approach for occupancy world model.

To motivate future work, we outline a few limitations based on our current comprehension: (1) The introduction of control modules to the base generative model increases computational complexity. Although our approach achieves superior performance than the baseline with lower overhead, further optimization to reduce computations remains valuable for real-time applications. (2) Our current multi-stage training pipeline could be streamlined. An end-to-end training scheme may improve efficiency while maintaining or enhancing model performance.

## Acknowledgments

This work was supported in part by the National Natural Science Foundation of China(52394264, 52372414,52472449), Beijing Natural Science Foundation(23L10038,L231008), and Beijing Municipal Science and Technology Commission (Z241100003524013, Z241100003524009).This work was also sponsored by Tsinghua University-DiDi Joint Research Center for Future Mobility and Tsinghua University-Toyota Joint Research Center.

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

# A Technical Appendices and Supplementary Material

## A.1 Broader Impacts

Our work proposes a generative occupancy world model with scene-centric forecasting as control, achieving state-of-the-art occupancy generation performance under multiple settings. This advancement leads to safer understanding, forecasting, generation and simulation of the driving scenes, which are crucial for real-world applications of autonomous driving (AD). Consequently, the most significant positive impact of our work is the enhancement of safety in AD systems.

However, the biggest negative societal impact of this work, as with any component of AD systems, is the safety concern. Autonomous driving systems are directly related to human lives, and erroneous or hallucinating predictions or generation can lead to hazardous understanding of the driving environment. Therefore, increasing the accuracy of occupancy generation to address false generation and illusions will require substantial follow-up efforts.

## A.2 Licenses for involved assets

The project code is built on top of the codebase provided by DOME [4], which is subject to the Apache License Version 2.0. Our experiments are conducted on the Occ3D-nuScenes [22] which provides occupancy labels for the nuScenes dataset [3] and Occ3D-Waymo [22] which provides occupancy labels for the Waymo open dataset [21]. Occ3D is licensed under the MIT license, nuScenes is licensed under the CC BY-NC-SA 4.0 license and Waymo open dataset is licensed under the Apache License Version 2.0.

## A.3 Discussion on the Design Philosophy of External Occupancy ControlNet

In this paper, we use a direct concatenation method for pose and BEV layout control, while we employ an external ControlNet for controlling scene-centric forecasting. We propose some explanations of the design philosophy.

Most importantly, We hope to remain the original generation capabilities of the occupancy world model unchanged. ControlNet only provides additional guidance to enhance the original world model, while direct concatenation makes the scene-centric forecasting an indispensable component for generation. This will downgrade the capability of the world model to a completion or inpainting model that inpaints future invisible voxels with the context of visible voxels. On the other hand, the model easily learns shortcut that outputs forecasting results directly on visible voxels. This shortcut learning may greatly hurt the multi-modal nature of both the generative model and the forecasting task. In contrast, ControlNet may be added to certain regions of the space or certain frames on the sequences, with user-defined masks and enjoy greater flexibility.

Moreover, poses (in the form of waypoints $[x, y, yaw]$) and bev layouts (in the form of semantic maps) are low-dimension conditions that empirically fit direct concatenation while occupancy sequences latents are high-dimension conditions that empirically fit external ControlNet.

## A.4 Discussion on Evaluation Metrics under End-to-end Planning Setting

Early autoregressive world models[35, 26] generates both future waypoints and future occupancy sequences at the predicted waypoints' coordinate at the same time. The evaluation between generated occupancies and ground-truth occupancies couples the similarity between the planning trajectory and the expert trajectory, and the similarity of generation. Our ablation experiment illustrates that both the translation error of the trajectory and the yaw angle error of the trajectory can greatly reduce the final IoU metric.

In order to factor out the influence of trajectory quality from the generation evaluation, we propose to reset the origin of the ground-truth occupancy according to the rotation matrix of the planning trajectory, and then evaluate the difference between the generated occupation and the ground-truth occupation. In this setting, we demonstrate that the generation performs similarly well when conditioned by the planning trajectory.

## A.5 More Implementation Details

### A.5.1 Environment Setup

The proposed algorithm runs in the python3.9 and torch2.5.1 environment and is expected to be compatible with the torch2.x environment. The environment needs to have mmcv 2.x and mmdet3d 1.1.x installed, and it is basically the same as the environment configuration scheme of OccWorld[35] and DOME[4]. We use AdamW optimizer for all experiments.

---

https://github.com/gusongen/DOME

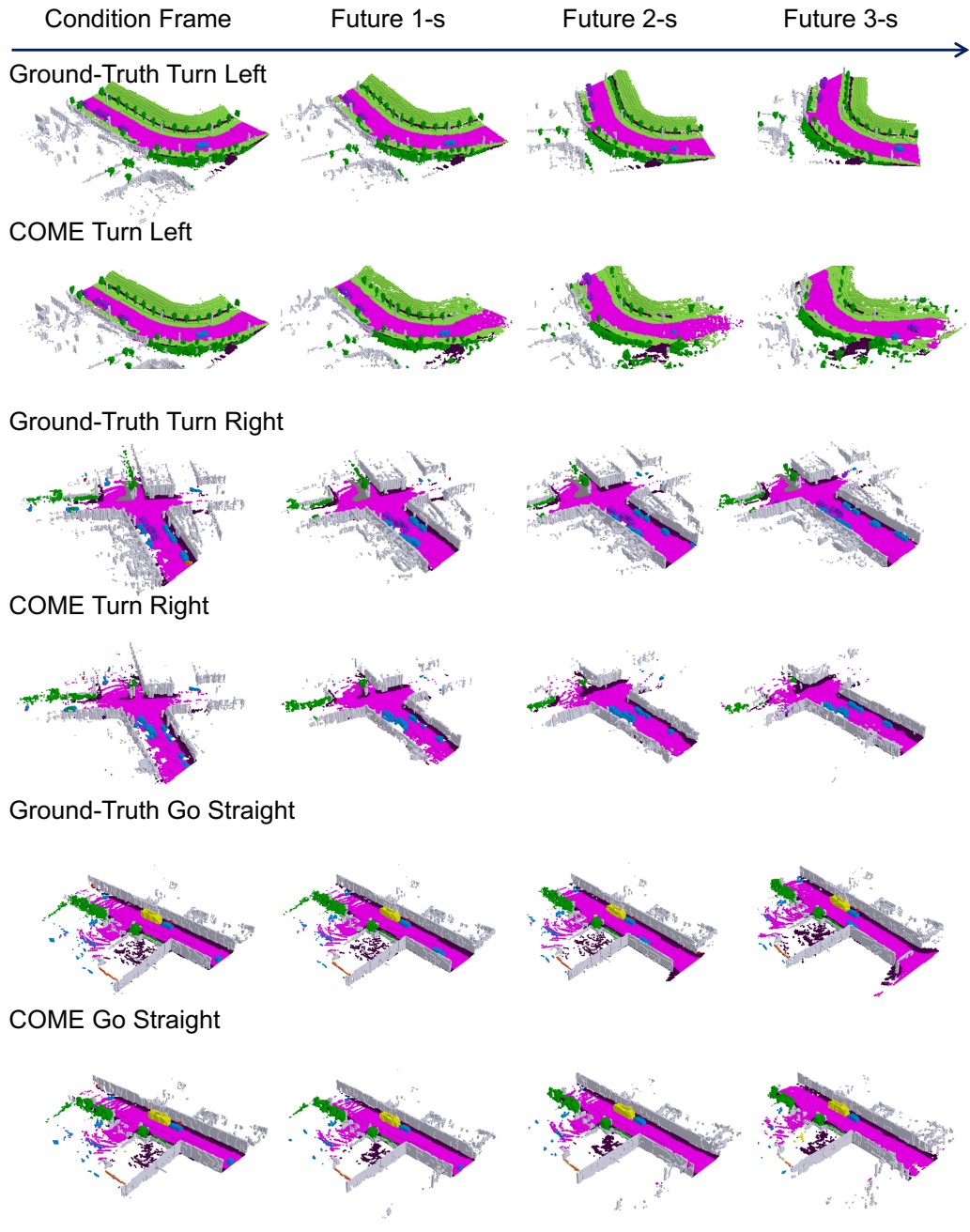

Figure 4: Visualization examples demonstrate the pose control alignment ability of COME generation. For different driving commands such as Go Straight, Turn Left and Turn Right, COME well follows the pose control and generate similar scenarios compared to ground-truth.

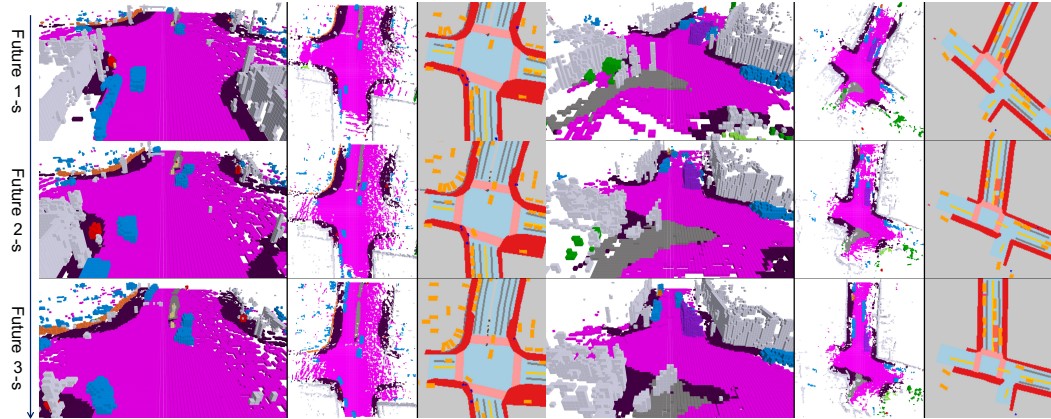

Figure 5: Visualization examples of occupancy generation with BEV layouts. COME generates occupancy sequences that well follows the BEV layout control.

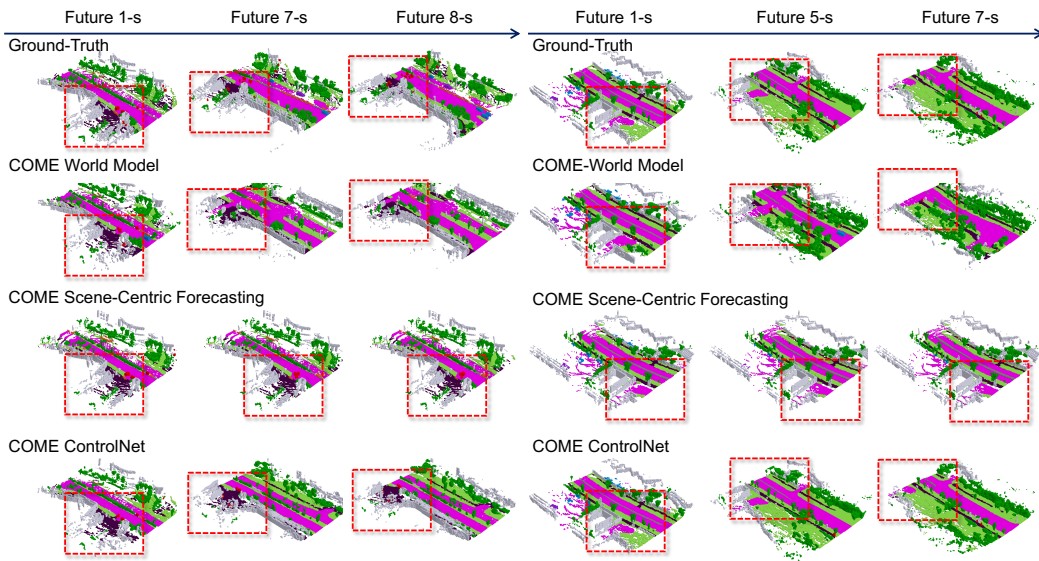

Figure 6: Visualization examples demonstrate the spatiotemporal consistency of differnet stages in COME for generating static environments. During the 8-second long generation process, COME world model mistakenly links drivable areas as intersections with the main street, but under the guidance of scene-centric forecasting, COME maintains the background consistency of road structures during generation.

Condition Frame     Future 1-s     Future 2-s     Future 3-s

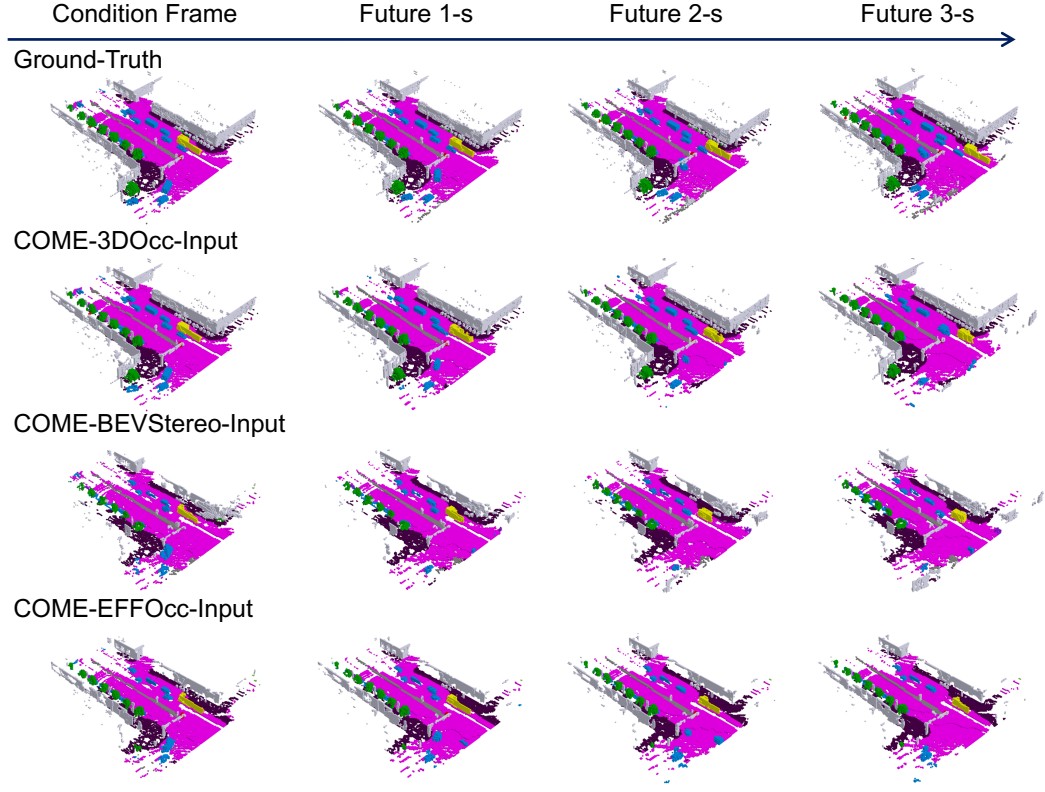

Figure 7: Visualization examples that takes 3D-Occ, vision-based BEVDet and fusion-based EFFOcc as occupancy sequences input.

Condition Frame     Future 1-s     Future 2-s     Future 3-s

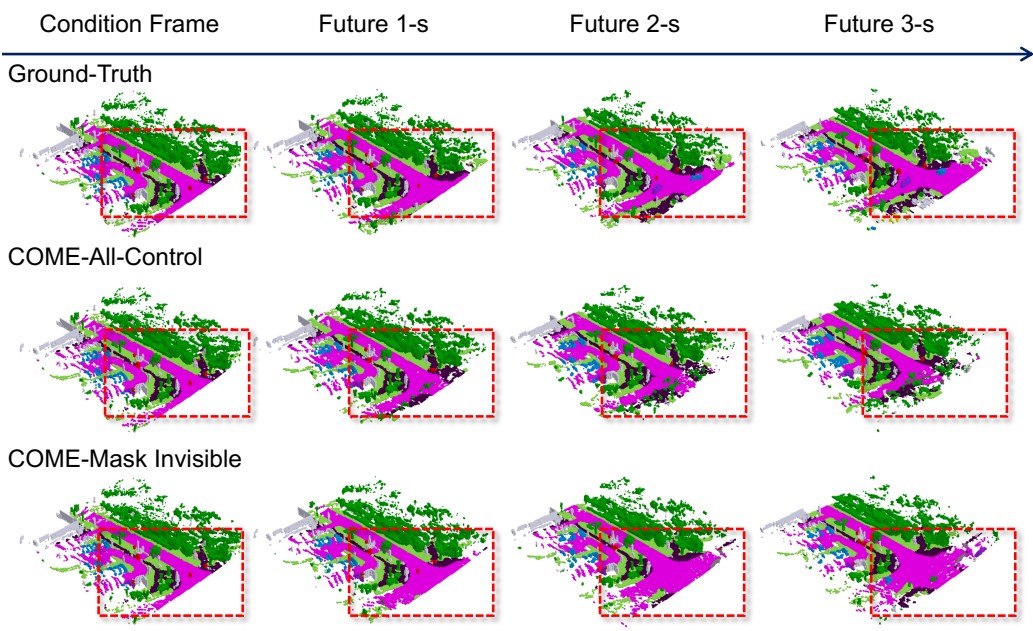

Figure 8: Visualization examples that uses different masking strategies during training and inference. Models with invisibility masks generally achieve much better qualitative results results.

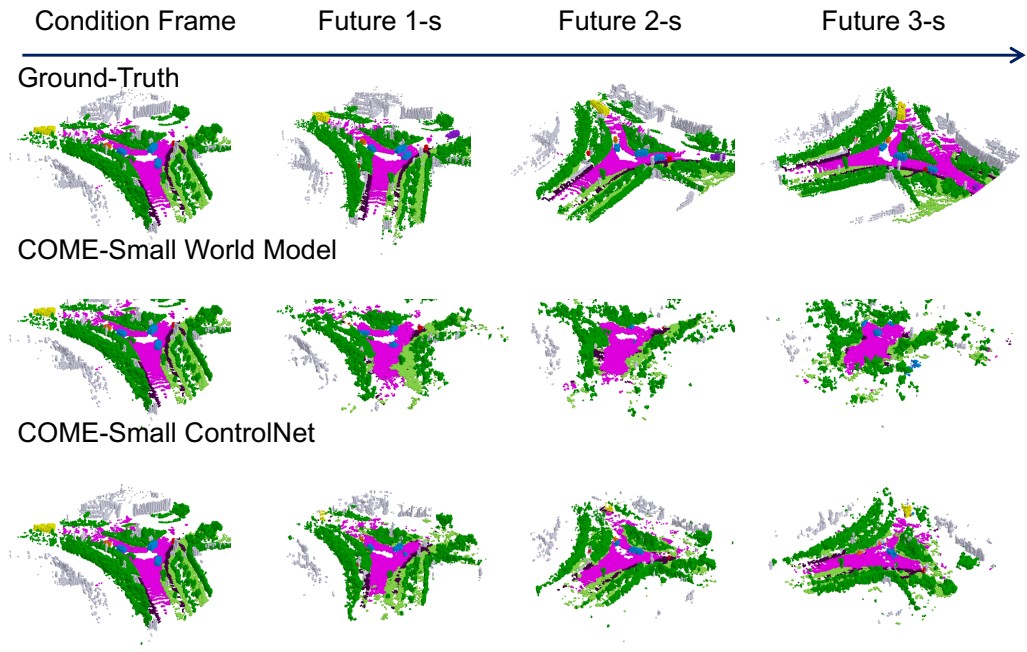

Figure 9: Visualization examples that uses small models. In this right turning example, the small model shows accurate pose control and good scene completion results.

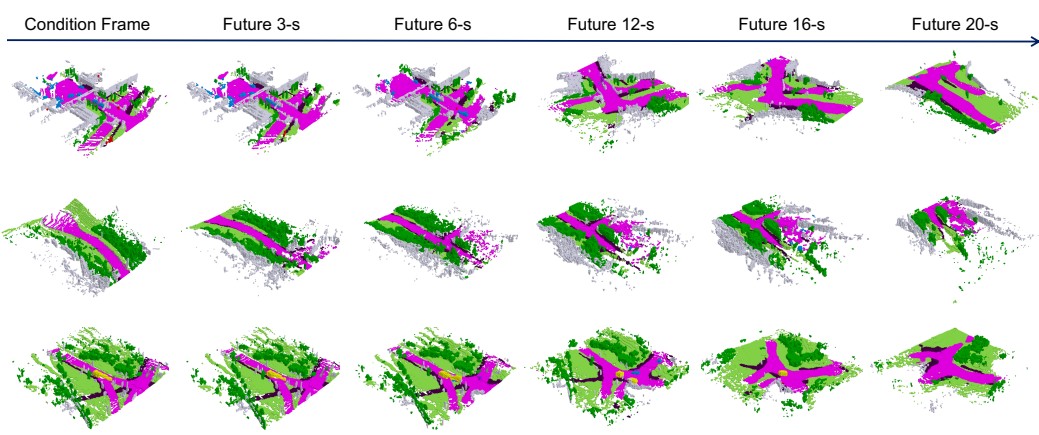

Figure 10: Visualization examples that uses repeated roll-out to generate super-long scenarios (20 second) with free trajectories.

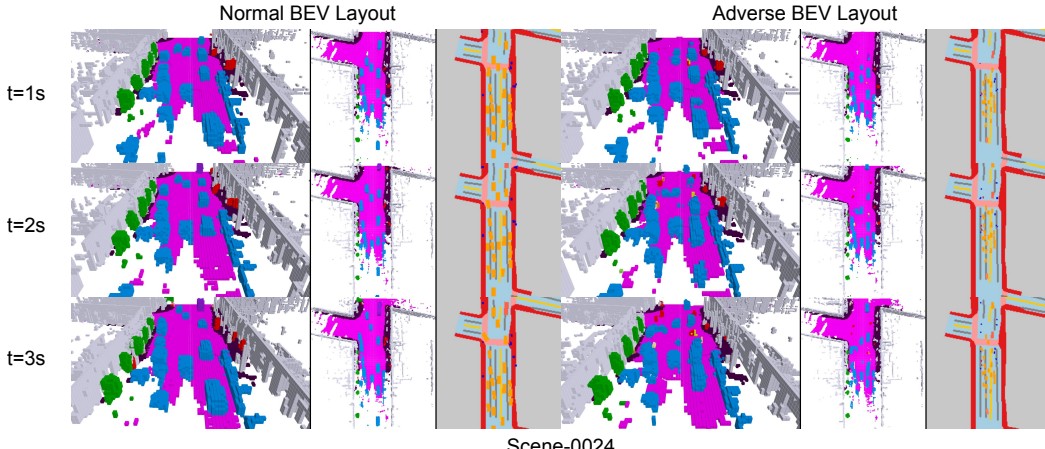

Figure 11: Visualization examples that uses adverse BEV layouts as inputs. In adverse BEV condition input (shown in right), all objects are re-placed to the centers which are half of the distance of the original centers of normal BEV layouts (shown in left).

### A.5.2 Networks Setup

**Continuous VAE** We use the same VAE with DOME[4] which consists of a 2D encoder and a 3D decoder. The class embedder has the expansion of $8$. The VAE has the compression ratio of $64$. The encoder downsampling and decoder upsampling ratios are set as $[1, 2, 4, 8]$. The base channel is set as $64$. The number of residual blocks is $2$. The attention resolution is $50$. The shapes of intermediate feature maps are $[[200, 200], [100, 100], [50, 50], [25, 25]]$.

**Scene-centric forecasting** The class embedder has the expansion of $32$. We use UNet with five-stage encoder and decoder. The encoder downsampling and decoder upsampling ratios are set as $[1, 2, 4, 8, 16]$ and the channel dimensions are $256, 512, 1024, 1024, 1024$ for each downsampling stride. The base channel is set as $256$. The number of temporal convolutional layers is set as $2$. We use InterpConv for upsampling operations.

**COME world model and ControlNet** We have two model sizes. By default, we use the model size similar to DiT-XL. The number of attention head is set as $12$, the hidden size is set as $768$. The number of layers (also named depth of world model) is set as $28$. We also set a smaller model. The number of attention head is set as $6$, the hidden size is set as $384$. The number of layers is set as $12$. The mlp_ratio is set as $4$. The patch size is set as $1$. The topk is set as $10$. For different size of world models, corresponding controlnet has half of the depth of the world model and uses the same parameters in each block.

**Network statistics.** The following statistics is tested with the standard task of generation future three second occupancy sequences with four-frame occupancy history. The scene-centric forecasting module has $27.31$ M parameters and $737.76$ GFLOPs. The base world model has $362.31$M parameters and $1375.38$ GLOPS. The base ControlNet has $158.49$M parameters and $691.23$ GLOPS. The base world model has $45.83$M parameters and $147.85$ GLOPS. The base ControlNet has $17.27$M parameters and $74.86$ GLOPS.

### A.5.3 Running Parameters Setup

**Diffusion parameters.** We use DDPM[5] as the default diffuser. By default, we use $1000$ denoising steps for training and $20$ denoising steps for inference. We also find in the ablation that $10$ denoising steps for inference only very slightly drops the final performance. The guidance scale is set as $7.5$. If the historical frame number is $4$, the conditional frames in training may be $[], [0], [0, 1], [0, 1, 2], [0, 1, 2, 3]$ and the conditional frames in inference is $[0, 1, 2, 3]$. The possibility of using pose as condition in training is set as $0.9$.

### A.5.4 Optional Inputs

**Sensor Inputs.** For vision track, we use the officially released checkpoint of BEVDet[7], BEVStereo, with Swin Transformer[16] base and image size $512 \times 1408$ as input and $1$ historical frames, achieves 3D occupancy prediction mIou of $42.45$. For LiDAR-camera fusion track, we use the officially released checkpoint of EFFOcc[20]. EFFOcc, with Swin Transformer[16] base and image size $512 \times 1408$ as input, achieves 3D occupancy prediction mIou of $54.08$. As 3D occupancy models are trained with camera mask on nuScenes dataset, we also use camera mask during occupancy generation evalaution.

Table 7: Results on COME with different control masking strategies.

| Masking Strategy | mIoU (%) ↑ | | | | IoU (%) ↑ | | | |
|---|---|---|---|---|---|---|---|---|
| | 1s | 2s | 3s | Avg. | 1s | 2s | 3s | Avg. |
| No Mask | 38.33 | 35.05 | 29.90 | 36.19 | 51.67 | 45.32 | 40.90 | 45.97 |
| Mask Condition | 43.25 | 34.05 | 28.82 | 35.37 | 50.86 | 43.78 | 39.05 | 44.56 |
| Random Dropout | 40.32 | 30.55 | 36.69 | 31.97 | 48.78 | 41.78 | 25.04 | 42.42 |
| Mask Control | 42.75 | 32.97 | 26.98 | 34.23 | 50.57 | 43.47 | 38.36 | 44.13 |

**Planning trajectory Input** COME does not strongly bind the world model to planning, instead, COME is controlled by the ego future trajectory as input. To compare with other world models with planning, we train a unsupervised simple imitation learning planning framework built upon BEV-Planner[14]. The planning module takes multi-view images as sensor input and outputs six waypoints including 2d translation and relative yaw angle compared to the current frame. The only supervision is the expert trajectory with ground-truth yaw angles. The planning module has an 3-s average L2 error of $0.48m$ under BEV-Planner open-loop metric.

The vast majority of planning algorithms are centered around the current coordinate of the ego vehicle, without including past or future perspectives, so more complicated planning modules can be placed as downstream or a parallel heads which integrates with the scene-centric forecasting module. In this way, planning trajectories can be integrated as control conditions into the generative model and theControlNet. We leave better planning modules considering scene-centric forecasting as future research.

**BEV Layout Input** The pre-processing for BEV layouts are the same as UniScene[11]. 3D bounding boxes of annotated objects are splatted to the BEV plane. The static layouts are computed with polygons of the high-definition maps.

## A.6    More Quantitative Results

### A.6.1    Generation with Different Masking Strategies

Tab. 7 demonstrates how masking strategies affect the final quantitative results. We find that model without any masking performs the best mIoU and IoU metrics, but it has a strong tendency to generate invisible areas as free. On the other hand, the model with masked control has much better visualization performance, but the quantitative results are lower than model without masking. In the main paper, we use model with masked control by default.

We find the possible reason for the formation of a trend that opposes quantitative results and qualitative results, is that the deductions for incorrect generation of invisible areas (generated content that does not match the true value) are higher for non-generation of invisible areas (where all areas that need to be generated are left blank) under the existing metrics.

We also try with different masking strategies. We try to perform masks on conditions before COME ControlNet rather than masks on controls after COME ControlNet, but the qualitative results are poor, so we do not pose the masks in condition features. We test with random dropout of masking in training stage, this operation helps better visualization quality but decreases the quantitative results. Finally, we find the best practice to balance quantitative results and qualitative results, that is to use a fixed invisibility mask on control features after COME ControlNet both in training and inference stages. As a result, we report this model and its variants in the main paper.

## A.7    More Visualizations

We show more visualization results including the generation results with different driving commands and future trajectoriesFig. 4, with BEV layouts as input (Fig. 5), comparison between results across stages (Fig. 6), comparison between different sensor inputs (Fig. 7), comparison between different masking strategies (Fig. 8), visualization with small models (Fig. 9), super-long occupancy video generation (Fig. 10), generation with costumed adverse BEV layouts (Fig. 11).

Fig. 4 demonstrates that COME can well align occupancy generation results with different driving commands (turn left, turn right, go straight) and pose control. The accurate pose control credits to the explicit transformation modelling guidance from scene-centric forecasting.

Fig. 5 demonstrates that COME can well align occupancy generation results with BEV layouts as condition.

Fig. 6 demonstrates an example of how scene-centric forecasting helps ego-centric generation. With COME world model ego-centric generation alone, after a long generation time, a new path appears at the roadside where

there is originally a separating strip, forming an intersection, which is inconsistent with previous observations. The scene-centric forecasting easily learns the static nature of the scenario. With scene-centric forecasting as control, COME ControlNet generates correct road structures and avoid the inconsistency caused by COME world model.

Fig. 7 demonstrates that the generation from sensor inputs are similarly well for the same scene. When the 3D occupancy quality is better, the generation results are also better.

Fig. 8 compares the generation results with different masking strategies. If we add full-space control regions, the model has an increasing tendency of generation invisible areas as free. If we mask control features on invisible areas (COME-Mask Invisible), the generation results are more satisfactory on most cases. We find that the performance of the visualization is not always consistent with the quality of the metric, which may be due to the fact that the metric was too restrictive. Specifically, when we mask the control of unseen areas, it is very reasonable to have free generation of unseen areas, but the reported metrics decrease because it is very likely that free generation is not the same as the truth value. If we exert control over the whole world, there is a higher tendency to set the unseen area to free, which in turn improves the quantitative results.

Fig. 9 shows an example that the smaller model achieves equally good generation results with ControlNet. The ego vehicle is turning at a large angle. The small world model has limited generation quality. However, with ControlNet, the small COME demonstrates better pose control and satisfactory generation results.

Fig. 10 shows some examples that the model roll-outs several times to generate super-long videos. This roll-out mechanism is similar to DOME[4] except that we use COME ControlNet for the first roll and use COME world model without ControlNet for the next rolls. This shows the fleixibility of our framework with and without ControlNet guidance at the same time.

Fig. 11 shows the simulation ability of COME world model to adverse BEV layouts as inputs. In this case, we try to reduce the center of each object box to half distance and create an adverse busy intersection. The 3D occupancy sequences can accurately respond to the BEV layout condition. A potential issue is conflicts between the Scene-Centric Forecasting module and BEV-layout control conditions (e.g., other vehicles' behavior conflicting with historical inertia). We find an elegant fix, that is to mark adversarial simulation-focused agent areas, mask the corresponding Scene-Centric Forecasting results, ensuring only BEV-layout covers these regions to maintain global physical consistency and local customization.

