# OpenReview forum: "COME: Adding Scene-Centric Forecasting Control to Occupancy World Model"
_NeurIPS.cc/2025/Conference — NeurIPS 2025 poster_

### Official Review · Reviewer_Gif3 · 2025-06-22

**Clarity:** 3
**Significance:** 3
**Originality:** 3
**Rating:** 4
**Confidence:** 3

**Summary:**

This paper proposes a world model for 3D occupancy forecasting.The key idea is to disentangle the evolution of the scene from that of the ego-vehicle motion. The method consists of three parts: 1) scene-centric forecasting using a unet architecture on historical occupancy. 2) World-model inference predicting future occupancy conditioned on historical occupancy, trajectory, or BEV layout. The paper demonstrates strong performance on the nuScenes 3D-Occ dataset.

**Questions:**

Is it possible to verify that the method works on more diverse driving scenarios?
How does the model handle user-specified changes to the BEV-layout?
How does the model handle adversarial scenarios or scenarios where a crash is imminent?

**Ethical Concerns:**

["NO or VERY MINOR ethics concerns only"]

**Final Justification:**

The authors resolved the concerns I had and I think the updated paper will be a good contribution to the community.

**Limitations:**

yes

**Quality:**

3

**Strengths And Weaknesses:**

_Strengths_:
The idea of separating ego-motion from scene evolution is elegant and intuitive, and the authors demonstrate that this design choice leads to improved forecasting performance. I think the method is well-motivated and clearly presented. The paper is generally well-written and easy to follow. The experimental results are strong, with competitive performance on nuScenes 3D occupancy prediction benchmarks. Also, the ablation studies are fairly informative and help validate the contributions of the method’s components.

_Weaknesses_:
The method is only evaluated on nuScenes, which is relatively constrained in terms of driving diversity. Ego-motion is often limited to driving in-lane following the road, and the environment contains few truly complex or rare events. This limits insight into how the model might generalise to more diverse or interactive settings. Furthermore, the paper does not explore the generation of fictional or counterfactual scenarios, which is a key promise of using world models for training and evaluation data. For example, it is unclear whether the model could simulate a crash scenario or how it would react to structural changes in the environment (e.g., the insertion or removal of a lane or a pedestrian crossing). It would strengthen the paper to evaluate or at least qualitatively discuss the model's capacity for reasoning about interventions or rare events, which are crucial for planning and decision-making.

---

> ### Author Rebuttal · Authors · 2025-07-31
>
> We thank the reviewer for recognition of novelty, writing and experimental results. We sincerely thank the reviewer for suggesting evaluating the proposed method under more diverse driving scenarios and adversarial conditions. We have supplemented initial verification on Occ3D-Waymo\[1] and are currently working with the new dataset. Additionally, we have added analyses explaining how to ensure generated results closely align with input condition changes to meet simulation needs.
>
> * **\[W1]**: The method is only evaluated on nuScenes, which is relatively constrained in terms of driving diversity. Ego-motion is often limited to driving in-lane following the road, and the environment contains few truly complex or rare events. This limits insight into how the model might generalise to more diverse or interactive settings.&#x20;
>
> **\[A1]**: nuScenes has limited diversity in ego-motion and complex objects, so we add experiments on the more complex Waymo Open Dataset (WOD). Occ3d-Waymo, built on WOD perception data, includes 798 training and 202 test sequences—richer and more diverse than nuScenes’ training scenes—with 10Hz sensor data and annotations. Our task is the same as nuScenes: predict 3s of future using 2s of past occupancy data at 0.5s intervals.
>
> We retrain OCC-VAE, UNet, DOME \[2], COME world model, and COME ControlNet on Occ3d-Waymo, initializing from nuScenes-trained components to save cost and time, with unchanged model sizes and training schedules.
>
> Currently we report a COME variant with the same structure and with stronger geometric controls from the scene-centric forecasting module, differing by additionally replacing noisy BEV latents with UNet-encoded latents in future visible areas during denoising. This variant converges faster, yields slightly better quantitative results but risks reducing prediction multimodality. Due to computational limits, the default COME model requires a few more training days (Current status: mIoU=29.35 and IoU=46.46 in epoch 20/1000,  already surpassing DOME and COME world model).
>
> VAE reconstruction quality: mIoU=74.88, IoU=82.57.
>
> The results of Waymo Occ3D are shown in the following tables. In general comparison (Table. 13) , our reproduced DOME achieves similar performance as in nuScenes dataset. COME world model has marginally better performance with DOME by 2.38 mIoU and 0.32 IoU. Scene-centric forecasting module outperforms the generative world model by a large margin (63% mIoU and 19.5% IoU). COME ControlNet, which built on control features provided with UNet module, maintains the same 3-s average mIoU and further improves 3-s average IoU to 23.7% improvement.
>
> | Method                                  | 1s-mIoU | 2s-mIoU | 3s-mIoU | 3s-AVG mIoU | 1s-IoU | 2s-IoU | 3s-IoU | 3s-AVG IoU |
> | --------------------------------------- | ------- | ------- | ------- | ----------- | ------ | ------ | ------ | ---------- |
> | DOME                                    | 31.62   | 24.82   | 19.65   | 25.36       | 51.24  | 43.17  | 37.47  | 43.96      |
> | COME Stage 1: World Model               | 34.45   | 26.28   | 22.50   | 27.74       | 51.16  | 43.24  | 38.44  | 44.28      |
> | COME Stage 2: Scene-Centric Forecasting | 47.37   | 40.85   | 36.31   | 41.51       | 59.69  | 52.27  | 45.60  | 52.52      |
> | COME Stage 3: ControlNet                | 47.29   | 40.80   | 36.44   | 41.51       | 60.63  | 53.98  | 48.62  | 54.41      |
>
> Table 13. Comparisons of general 4D occupancy forecasting performance between DOME and different stages of COME on Waymo Occ3D dataset.&#x20;
>
>
>
> We also compare the per-category 4D-occupancy forecasting performance of moving objects in Waymo Occ3D dataset in Table. 14. COME outperforms DOME on all moving categories by a large margin.&#x20;
>
> | Method | Vehicle IoU | Bicyclist IoU | Pedestrain IoU | Bicycle mIoU | Motorcycle-IoU |
> | ------ | ----------- | ------------- | -------------- | ------------ | -------------- |
> | DOME   | 38.28       | 16.83         | 21.15          | 24.34        | 9.9            |
> | COME   | 50.98       | 31.85         | 41.63          | 36.78        | 27.75          |
>
> Table 14. Comparisons of 4D occupancy forecasting performance of moving categories between DOME and COME on Waymo Occ3D dataset.&#x20;
>
> To demonstrate the prediction of truly complex or rare events, we hope to further train and test our method on industrial-level private autonomous driving data and corner cases.
>
> * **\[W2]**: Furthermore, the paper does not explore the generation of fictional or counterfactual scenarios, which is a key promise of using world models for training and evaluation data. For example, it is unclear whether the model could simulate a crash scenario or how it would react to structural changes in the environment (e.g., the insertion or removal of a lane or a pedestrian crossing). It would strengthen the paper to evaluate or at least qualitatively discuss the model's capacity for reasoning about interventions or rare events, which are crucial for planning and decision-making.
>
>
>
> **\[A2]**: Thanks for pointing out the potential application of COME. World models for simulation typically use obstacle box positions and maps as control conditions, generating content that adapts to changes in these conditions. Our BEV Condition control scheme mirrors UniScene \[3], which yields good rendering with customized BEV layout. Unlike UniScene, our method adds external controls for physics-consistent prediction, with adaptive scene-centric forecasting control (details in A3.2).
>
>
>
> We test static (by randomly removing half map patches) and dynamic (by halving box-to-car center distances) scene distortions. In both, COME prioritizes BEV layout guidance rather than conditions from the scene-centric forecasting module, generating occupancy maps qualitatively matching the layouts. Per new NeurIPS guidelines, results can’t be visualized but will be added to supplements in revision.
>
>
>
> * **\[Q1]**: **\[Q1.1]: Is it possible to verify that the method works on more diverse driving scenarios? **\[Q1.2]: How does the model handle user-specified changes to the BEV-layout? **\[Q1.3]: How does the model handle adversarial scenarios or scenarios where a crash is imminent?
>
>
>
> **\[A3.1]**: We verify the method on the Waymo Open Dataset, which has more diverse driving scenarios.
>
> **\[A3.2]**: The model accepts user-specified changes or adversarial scenarios via BEV-layout modifications, validated in UniScenes. A potential issue is conflicts between the Scene-Centric Forecasting module and BEV-layout control conditions (e.g., other vehicles' behavior conflicting with historical inertia). An elegant fix: mark adversarial simulation-focused agent areas, mask the corresponding Scene-Centric Forecasting results, ensuring only BEV-layout covers these regions to maintain global physical consistency and local customization.
>
> **\[A3.3]**: COME's simulation model takes adversarial BEV layouts (e.g., from adversial traffic simulators like Challenger \[4]) as input conditions. When agents crash, COME merges them into a single occupancy instance. World models for recognizing and predicting such instances are left for future work.
>
>
>
> \[1] Occ3D: A Large-Scale 3D Occupancy Prediction Benchmark for Autonomous Driving, NeurIPS 2023.
>
> \[2] Dome: Taming diffusion model into high-fidelity controllable occupancy world model, arxiv 2024.
>
> \[3] Uniscene: Unified occupancy-centric driving scene generation, CVPR 2025.
>
> \[4] Challenger: Affordable Adversarial Driving Video Generation

---

> > ### Comment · Reviewer_Gif3 · 2025-08-05
> >
> > Thank you for the rebuttal. You have resolved the concerns I had and I think the updated paper will be a good contribution to the community.

---

> > > ### Author Response · Authors · 2025-08-08
> > >
> > > Thank you very much for your thoughtful review and valuable suggestions for our work. We truly appreciate the time and care in reading our responses and experiments. We believe your comments have directly contributed to improving the completeness of our paper, especially the practical applications of occupancy world models.

---

### Official Review · Reviewer_hsxA · 2025-06-24

**Clarity:** 2
**Significance:** 2
**Originality:** 2
**Rating:** 4
**Confidence:** 3

**Summary:**

This paper proposes a simple scene-wise forecasting module by leveraging the observation that a significant portion of voxels in the occupancy world model are static and shared across sequential scenes. The output of this module is then used as a condition for the COME ControlNet, resulting in improved performance. The proposed approach is evaluated through comparisons of 4D occupancy generation performance using various input modalities, including camera inputs and occupancy inputs.

**Questions:**

1. L158 and L160: Definitions for D_t and D_τ are missing. -> This creates confusion regarding how the transformed sequence x' is handled. Are all sequential voxels concatenated into x' ?
2. Would directly using the transformed voxels as conditions yield a similar effect? This approach may be analogous to the use of warping in the image domain to leverage information from previous frames, as shown in [1].
[1] Streetscapes: Large-scale Consistent Street View Generation Using Autoregressive Video Diffusion

**Ethical Concerns:**

["NO or VERY MINOR ethics concerns only"]

**Final Justification:**

All of my concerns have been adequately addressed.

**Limitations:**

yes

**Paper Formatting Concerns:**

There are no major formatting issues in the paper.

**Quality:**

2

**Strengths And Weaknesses:**

[Strengths]
1. This paper proposes a method that leverages the consistency of scene-centric representations to construct scene conditions.
2. The proposed method demonstrates significant performance improvements over the baselines across various input settings.

[Weaknesses]
1. In Table 1, when comparing methods that utilize camera input, the same occupancy prediction model should be used across all baselines for a fair comparison. To properly attribute the performance improvement to the proposed method, a fair comparison using an identical occupancy prediction backbone is necessary.
2. Given that the scene-centric forecasting module relies on a large proportion of static voxels. In that case, what is the specific advantage of using predicted future occupancy prediction from a simple Unet architecture as a condition?
3. Since the output of the scene-centric forecasting module is used as a condition, concerns arise about how moving objects are accounted for. In a world model, rather than simply maintaining the static regions of a scene, it is important to predict how the scene will evolve. It is unclear how the proposed method addresses this aspect.

---

> ### Author Rebuttal · Authors · 2025-07-31
>
> We thank the reviewer for recognition of experimental results. We sincerely thank the reviewer for suggestions to enhance originality clarification. We supplement comparison experiments under identical camera and ego trajectory inputs, experiments with varying input conditions to demonstrate the effectiveness of scene-centric forecasting for downstream generation, and statistics on dynamic categories.
>
> We respectfully disagree with originality. Prior works (DOME \[10], OccWorld \[1]) remain in ego-centric coordinates, suffering performance drop under perspective shifts. Our novelty lies in recognizing that \*\*short-term scene-centric forecasting of moving regions is sufficient and necessary\*\*; we are the first to formalize this as a ControlNet condition. The ablation in Table 10 shows that removing this component leads to a 31 % performance drop, confirming our nontrivial contribution and novelty. Below are our responses to other specific concerns:
>
> * **\[W1]**: The same occupancy prediction model with camera input should be used across all baselines for a fair comparison.
>
> **\[A1]**: Thank you for highlighting the need to address experiments with potentially different setups and inputs. We prepare two occupancy predictors that are the same as prior works and demonstrate COME's performance in Table 8. COME with both inputs receives the best performances.
>
> | Model              | Occupancy Predictor           | Image Backbone        | 3s-AVG mIoU | 3s-AVG IoU |
> | ------------------ | ----------------------------- | --------------------- | ----------- | ---------- |
> | OccWorld \[1]      | TPVFormer \[2]                | N/A                   | 8.62        | 16.53      |
> | OccLLaMA \[3]      | FB-OCC \[4]                   | N/A                   | 8.66        | 22.99      |
> | OccVAR \[5]        | FB-OCC                        | N/A                   | 11.79       | 24.35      |
> | Occ-LLM \[7]       | FB-OCC                        | N/A                   | 10.21       | 23.79      |
> | COME (Ours)        | FB-OCC                        |                   ResNet-50 | 16.09       | 33.02      |
> | DFIT-OccWorld \[6] | BEVDet Series (Not Specified) | N/A                   | 10.50       | 17.02      |
> | RenderWorld \[8]   | BEVStereo \[9]                | Swin-B                | 2.58        | 13.73      |
> | COME (Ours)        | BEVStereo                     | Swin-B                | 19.11       | 37.63      |
>
> Table 8. Comparisons of Camera-input occupancy world models
>
> We add a quantitative comparison between DOME and COME with strictly the same input, as shown in Table. 9. Compared to DOME, COME has 17.5% mIoU and 8.5% improvement under BEV-Planner\[11] ego trajectories, and 17.3% mIoU and 15.7% IoU improvement under ground-truth ego trajectories. &#x20;
>
> | Model | Occupancy Predictor | Image Backbone | Ego Trajectory | 3s-AVG mIoU | 3s-AVG IoU |
> | ----- | ------------------- | -------------- | -------------- | ----------- | ---------- |
> | DOME  | BEVStereo           | Swin-B         | BEV-Planner    | 16.15       | 34.68      |
> | COME  | BEVStereo           | Swin-B         | BEV-Planner    | 19.11       | 37.63      |
> | DOME  | BEVStereo           | Swin-B         | GT             | 18.97       | 38.17      |
> | COME  | BEVStereo           | Swin-B         | GT             | 22.26       | 44.07      |
>
> Table 9. Comparison with aligned camera input and ego trajectory.
>
> * **\[W2]**: Given many static voxels, what is the specific advantage of using predicted future occupancy prediction from a simple Unet architecture as a condition?
>
> **\[A2]**: The scene-centric forecasting module handles not only static voxels but also dynamic agents, retaining static voxels while accurately predicting future dynamic voxels. A key advantage is that pre-processing rigid body coordinate transformations simplify predicting from ego-centric to scene-centric views. Results for dynamic categories (Table. 11) demonstrate the module’s prediction ability.
>
> We add an experiment (Table 10) where the COME world model is controlled solely by static voxels. This yielded marginal improvements (0.37 mIoU, 1.23 IoU), showing static voxels as controls are less effective than the proposed UNet module.
>
> | Model            | Condition     | 1s-mIoU | 2s-mIoU | 3s-mIoU | 3s-AVG mIoU | 1s-IoU | 2s-IoU | 3s-IoU | 3s-AVG IoU |
> | ---------------- | ------------- | ------- | ------- | ------- | ----------- | ------ | ------ | ------ | ---------- |
> | COME-World-Model | N/A           | 30.54   | 21.89   | 17.74   | 23.39       | 38.99  | 31.38  | 26.66  | 32.34      |
> | COME-ControlNet  | Static Voxels | 30.82   | 22.24   | 18.21   | 23.76       | 40.28  | 32.33  | 28.11  | 33.57      |
> | COME-ControlNet  | UNet          | 42.75   | 32.97   | 26.98   | 34.23       | 50.57  | 43.47  | 38.36  | 44.13      |
>
> Table 10. Model performance when COME world model is controlled by static voxels.
>
>
>
> * **\[W3]**: How moving objects are accounted for in a world model?
>
> **\[A3]**: The scene-centric forecasting module perceives static surroundings and strongly predicts moving objects’ evolution, providing a geometric prior for generative models. We compare moving object prediction across the stage-1 world model, stage-2 scene-centric forecasting, and stage-3 COME ControlNet, listing IoU metrics for moving categories below. Table 11 reports the average IoU of moving categories across 6 future frames (0.5s to 3.0s). The UNet module outperforms DOME in predicting all moving categories, so using UNet as a prior significantly enhances COME-ControlNet’s dynamic object prediction.
>
> | Model                          | Bicycle 3-s AVG IoU | Bus 3-s AVG IoU | Car 3-s AVG IoU | Construction Vehicle 3-s AVG IoU | Motorcycle 3-s AVG IoU | Pedestrian 3-s AVG IoU | Trailer3-s AVG IoU | Truck3-s AVG IoU |
> | ------------------------------ | ------------------- | --------------- | --------------- | -------------------------------- | ---------------------- | ---------------------- | ------------------ | ---------------- |
> | DOME                           | 24.15               | 24.76           | 27.17           | 29.15                            | 15.04                  | 11.81                  | 26.71              | 27.12            |
> | COME-World-Model               | 21.12               | 24.88           | 24.02           | 21.75                            | 10.84                  | 9.72                   | 26.16              | 24.65            |
> | COME-Scene-Centric Forecasting | 30.71               | 38.92           | 37.04           | 39.70                            | 30.03                  | 32.53                  | 41.43              | 42.89            |
> | COME-ControlNet                | 29.14               | 33.55           | 31.83           | 33.84                            | 28.34                  | 23.11                  | 34.49              | 30.75            |
>
> Table 11. Comparison of model performance on dynamic categories on nuScenes dataset.
>
> * **\[Q1]**: L158 and L160: Definitions for D\_t and D\_τ are missing. -> This creates confusion regarding how the transformed sequence x' is handled. Are all sequential voxels concatenated into x' ?
>
> **\[A4]**:  We will thoroughly rewrite the UNet section. D\_t denotes the number of input historical frames, and D\_τ the number of output future frames; both are set based on the task formulation. The transformation from x to x' is a simple rigid transformation: each voxel’s center point in x is converted to the current frame’s coordinates, resulting in x' having the same shape as x.
>
> * **\[Q2]**: Would directly using the transformed voxels as conditions yield a similar effect? This approach may be analogous to the use of warping in the image domain to leverage information from previous frames, as shown in \[1]. \[1] Streetscapes: Large-scale Consistent Street View Generation Using Autoregressive Video Diffusion
>
> **\[A5]**: We add an experiment that all visible voxels are transformed and then inputted to COME ControlNet as conditions. Table 12 shows this variant improves marginally over the COME world model (0.88 mIoU, 0.81 IoU) but is far less effective than the UNet module.
>
>
>
> | Model            | Condition          | 1s-mIoU | 2s-mIoU | 3s-mIoU | 3s-AVG mIoU | 1s-IoU | 2s-IoU | 3s-IoU | 3s-AVG IoU |
> | ---------------- | ------------------ | ------- | ------- | ------- | ----------- | ------ | ------ | ------ | ---------- |
> | COME-World-Model | N/A                | 30.54   | 21.89   | 17.74   | 23.39       | 38.99  | 31.38  | 26.66  | 32.34      |
> | COME-ControlNet  | Transformed Voxels | 31.31   | 22.66   | 18.85   | 24.27       | 39.82  | 31.86  | 27.76  | 33.15      |
> | COME-ControlNet  | UNet               | 42.75   | 32.97   | 26.98   | 34.23       | 50.57  | 43.47  | 38.36  | 44.13      |
>
> Table 12. Model performance when COME world model is controlled by all transformed voxels.
>
>
> \[1] OccWorld: Learning a 3D Occupancy World Model for Autonomous Driving, ECCV2024.
>
> \[2]Tri-Perspective View for Vision-Based 3D Semantic Occupancy Prediction, CVPR2023.
>
> \[3] OccLLaMA: An Occupancy-Language-Action Generative World Model for Autonomous Driving, Arxiv 2024.
>
> \[4] FB-OCC: 3D Occupancy Prediction based on Forward-Backward View Transformation, Arxiv 2023.
>
> \[5] OccVAR: Scalable 4D Occupancy Prediction via Next-Scale Prediction, Openreview 2024.
>
> \[6] An Efficient Occupancy World Model via Decoupled Dynamic Flow and Image-assisted Training, Arxiv 2024.
>
> \[7] Occ-LLM: Enhancing Autonomous Driving with Occupancy-Based Large Language Models, ICRA2025.
>
> \[8] RenderWorld: World Model with Self-Supervised 3D Label, ICRA2025.
>
> \[9] BEVStereo: Enhancing Depth Estimation in Multi-view 3D Object Detection with Dynamic Temporal Stereo, AAAI2023.
>
> \[10] Dome: Taming diffusion model into high-fidelity controllable occupancy world model, arxiv 2024.
>
> \[11] Is Ego Status All You Need for Open-Loop End-to-End Autonomous Driving, CVPR2024.

---

> > ### Author Response · Authors · 2025-08-08
> >
> > We thank the reviewer for the thoughtful thinking and valuable suggestions. As the discussion phase is approaching its end, we hope the above clarifications and the additional experiments sufficiently addressed your concerns. If you are satisfied, we kindly request you to consider updating the score to reflect the newly added results and discussion. We remain committed to addressing any remaining points you may have during the discussion phase.

---

> ### Comment · Reviewer_hsxA · 2025-08-08
>
> Thank you for the rebuttal. All of my concerns have been addressed. Therefore, I will raise my final rating.

---

### Official Review · Reviewer_exSF · 2025-07-02

**Clarity:** 2
**Significance:** 3
**Originality:** 3
**Rating:** 5
**Confidence:** 4

**Summary:**

Autonomous driving systems increasingly depend on accurate 3D world models to anticipate future scene dynamics and support safe decision-making. This paper introduces COME (Control into the Occupancy world ModEl), a novel approach for forecasting 3D occupancy maps in simulation. Unlike prior methods such as DOME that rely on ego-centric coordinates, COME adopts a scene-centric coordinate system, significantly reducing prediction errors from vehicle perspective shifts. The approach consists of three main components: (1) a diffusion-style transformer world model that predicts future occupancy maps conditioned on historical pose and auxiliary data, (2) a UNet-based forecasting module that restructures input data into scene-centric coordinates and forecasts future vehicle states, and (3) a ControlNet that injects scene-level conditions from the forecasting module into the world model for controllable prediction. On the nuScenes-Occ3D benchmark, COME demonstrates substantial performance improvements over existing methods, supported by extensive ablation studies and qualitative analysis. Overall, the innovative use of scene-centric coordinates and robust results make COME a significant contribution to the autonomous driving field.

**Questions:**

1. 161-163: Can you explicitly explain what inputs the scene-centric forecasting module receives and what it is predicting? Additionally, how are these predictions propagated to future timesteps? This section is not entirely clear.

**Ethical Concerns:**

["NO or VERY MINOR ethics concerns only"]

**Final Justification:**

The authors have done an admirable job addressing our concerns. Notably, they evaluated their method on an additional benchmark dataset, as requested by reviewers 2AyB and Gif3 — an effort I especially appreciate. They also responded thoroughly to my comments regarding clarity and motivation. Most importantly, they substantiated many of their claims with experimental results, reinforcing the advantages of COME over the baseline DOME. Overall, I find the proposed approach compelling and believe it will be valuable to the community, which is why I am recommending acceptance.

**Limitations:**

Yes

**Paper Formatting Concerns:**

None.

**Quality:**

3

**Strengths And Weaknesses:**

**Strengths**
1. Novel approach to scene occupancy synthesis for ego-centric sequence problems.
- The authors recognize that scene generation is significantly more difficult when using an ego-centric coordinate system, where small changes in vehicle orientation can dramatically alter the scene. Instead, they propose a scene-centric coordinate system, ensuring that perspective shifts from the ego-vehicle do not substantially impact the scene, which reduces the need for constant scene reconstruction as the vehicle moves.
- The method leverages ControlNet to inject control signals into the Occupancy World Model, facilitating scene synthesis over time. The key observation that 92.7% of occupied voxels are static means the model needs to update only a small portion of the scene. This insight, combined with ControlNet, reduces computational complexity for temporal updates.
- A masking operation is introduced to ignore control signals from ControlNet in regions not historically observed, since such areas tend to be noisy and difficult to predict accurately.
2. SOTA or near SOTA performance on multiple configurations on the Occ3D-nuScenes benchmark dataset for 4D occupancy generation.
- Performance gains shown in Table 1 are substantial, with IoU time averages exceeding the prior SOTA by 13.28%.
3. Comprehensive evaluation and ablation studies.
- Multiple useful experiments (including longer sequence forecasting up to 8s, masking strategies, and qualitative ablations) are included, even if some are in the appendix.
- The ablation study clarifies the importance of each component and demonstrates performance drops due to issues such as misalignment of 2D pose and yaw for predicted future waypoints.
- It also highlights the benefit of the ControlNet-style training protocol, emphasizing training one component at a time.

**Weaknesses**
1. Unclear motivation and applicability to real-world scenarios.
- The introduction lacks a clear explanation of why generating occupancy maps is critical for autonomous driving.
- More detail is needed on why perspective shifts from ego-centric systems are problematic, ideally with statistics or realistic examples.
2. Some lack of clarity and organization in the writing.
- 46-57: This section is difficult to follow, especially for readers unfamiliar with the methods. These details would fit better in the methods section.
- 58-61: Discussion of input sources and prediction horizons is misplaced in the introduction.
- 39-45, 66-68: Repeatedly foreshadowing SOTA results feels redundant.
- 153-163: The purpose and operation of the scene-centric forecasting UNet are not explained clearly enough.
3. Minor editing issues:
- 134: Periods for e.g.
- 543: Confusing sentence.

Overall, the method is interesting and delivers strong experimental results, but the motivation and presentation could be improved.

---

> ### Author Rebuttal · Authors · 2025-07-31
>
> We thank the reviewer for recoginition of novelty and experimental results. We sincerely thank the reviewer for pointing out clarification issues and suggesting potential improvements to the introduction and methodology sections. We have revised our draft accordingly to enhance its comprehensiveness. Below, please find our responses to other concerns:
>
>
>
> * **\[W1.1]**: Unclear motivation and applicability to real-world scenarios. The introduction lacks a clear explanation of why generating occupancy maps is critical for autonomous driving.
>
> **\[A1.1]**:&#x20; Thank you for pointing this out. We will add explanations in the revised draft. Occupancy is widely recognized as a key intermediate representation for understanding drivable space in autonomous driving, effectively handling spatial perception of general obstacles. The generative occupancy world model has multiple potential applications:
>
> (1) Predicting future occupancy grid maps based on planned trajectories and validating collision risks for the current trajectory \[1].
>
> (2) Serving as a critical intermediate result for synthesizing simulation data \[2]\[3]. Generated occupancies can be rendered to the image perspective via Gaussian splatting \[4], acting as accurate geometric controls for driving video generation.
>
>
>
>
>
> * **\[W1.2]**: More detail is needed on why perspective shifts from ego-centric systems are problematic, ideally with statistics or realistic examples.
>
> **\[A1.2]**:&#x20; Perspective shifts in ego-centric systems are problematic due to the challenge of relocating visible dynamic agents and static environments. We classify scenes as having perspective shifts based on whether the vehicle moved more than 5 meters in 5-second prediction clips. We test DOME \[5] and COME in these scenarios, with quantitative results in Table 7. DOME performs well in ego-static scenes (where other agents are static) but poorly in ego-dynamic ones, achieving only 47% mIoU and 52.9% IoU (vs. ego-static). In contrast, COME shows larger improvements in ego-dynamic scenes (36.3% mIoU, 26.88% IoU) than in ego-static ones (8.8% mIoU, 4.2% IoU), demonstrating the effectiveness of our scene-centric forecasting control for the world model.
>
>
>
> | Model | Scenarios | 1s-mIoU | 2s-mIoU | 3s-mIoU | 3s-AVG mIoU | 1s-IoU | 2s-IoU | 3s-IoU | 3s-AVG IoU |
> | ----- | --------- | ------- | ------- | ------- | ----------- | ------ | ------ | ------ | ---------- |
> | DOME  | All       | 35.11   | 25.89   | 20.29   | 27.10       | 43.99  | 35.36  | 29.74  | 36.36      |
> | DOME  | Static    | 56.89   | 47.73   | 40.20   | 48.27       | 67.44  | 61.29  | 53.56  | 60.76      |
> | DOME  | Dynamic   | 30.60   | 21.44   | 16.32   | 22.79       | 39.40  | 31.07  | 25.90  | 32.12      |
> | COME  | All       | 42.75   | 32.97   | 26.98   | 34.50       | 50.57  | 43.47  | 38.36  | 44.13      |
> | COME  | Static    | 61.27   | 52.32   | 45.27   | 52.95       | 68.66  | 64.15  | 57.80  | 63.53      |
> | COME  | Dynamic   | 39.15   | 29.60   | 24.46   | 31.06       | 47.05  | 39.82  | 35.42  | 40.76      |
>
> Table 7. Model performance on scenes classified according to ego movement.&#x20;
>
>
> * **\[W2]:** Some lack of clarity and organization in the writing: 46-57: This section is difficult to follow, especially for readers unfamiliar with the methods. These details would fit better in the methods section. 58-61: Discussion of input sources and prediction horizons is misplaced in the introduction. 39-45, 66-68: Repeatedly foreshadowing SOTA results feels redundant. 153-163: The purpose and operation of the scene-centric forecasting UNet are not explained clearly enough.
>
> **\[A2]:** Thanks for pointing out paragraphs that are hard to understand. We will revise the manuscript based on the instructions above. For l46-57, we will simplify the module introduction and place the original paragragh in methodology. For l58-61, we will place the test details in the experiment section. For l39-45, we will delete the redundant description. For l153-163, we will thoroughly rewrite this section to clarify its purpose and operation.
>
> * **\[W3]:** Minor editing issues: 134: Periods for e.g. 543: Confusing sentence.
>
> **\[A3]:** Thanks for pointing out editing errors, we will check and correct editing errors in the manuscript.  Line 543 should be revised as: We try to perform masks on conditions before COME ControlNet rather than masks on controls after COME ControlNet, but the qualitative results are poor, so we do not pose the masks in condition features.
>
>
> * **\[Q1]:** 161-163: Can you explicitly explain what inputs the scene-centric forecasting module receives and what it is predicting? Additionally, how are these predictions propagated to future timesteps? This section is not entirely clear.
>
> **\[A4]: &#x20;**&#x54;he scene-centric forecasting module takes historical occupancy sequences and ego poses as input. Historical occupancy sequences are transformed to the current ego pose, resulting in semantic occupancy sequences of shape \[D\_t, H, W, L] under the same coordinate system. The module outputs future scene-centric occupancy sequences of shape \[D\_τ, H, W, L], which are concatenated with the input sequences to form a tensor of shape \[D\_t + D\_τ, H, W, L]—serving as the input condition feature for COME ControlNet.
>
>
>
> \[1] Driving in the occupancy world: Vision-centric 4d occupancy forecasting and planning via world models for autonomous driving, AAAI 2025.
>
> \[2] Uniscene: Unified occupancy-centric driving scene generation, CVPR 2025.
>
> \[3] Drivingsphere: Building a high-fidelity 4d world for closed-loop simulation, CVPR2025.
>
> \[4] 3D Gaussian Splatting for Real-Time Radiance Field Rendering, ACM Transactions on Graphics (TOG) 2023.
>
> \[5] Dome: Taming diffusion model into high-fidelity controllable occupancy world model, arxiv 2024

---

### Official Review · Reviewer_2AyB · 2025-07-04

**Clarity:** 3
**Significance:** 3
**Originality:** 3
**Rating:** 4
**Confidence:** 5

**Summary:**

COME introduces a three-stage occupancy world‑model forecasting framework for autonomous driving: (1) it rigidly transforms ego‑centric historical occupancy grids into a fixed, scene‑centric coordinate frame and uses a lightweight UNet to predict short‑term future occupancy; (2) it inversely reprojects these predictions and encodes them into latent “scene‑condition” features; and (3) it injects those features via a ControlNet into a diffusion Transformer—filtered by a visibility mask—to produce long‑horizon occupancy forecasts with enhanced spatiotemporal consistency and controllability. On the nuScenes‑Occ3D benchmark, COME outperforms the state‑of‑the‑art DOME

**Questions:**

1)	Inference Latency & Throughput: What are the end‑to‑end inference times and achievable frame rates for 10, 20, and 40 denoising steps
2)	Visibility Mask Sensitivity: How sensitive are final forecasts to the visibility‑mask threshold \varepsilon ? Please provide an ablation study over a range of \varepsilon values.
3)	How is the performance of DOME in Tables 2, 3b and 4c? Is the conclusion consistent with COME ablation study?

**Ethical Concerns:**

["NO or VERY MINOR ethics concerns only"]

**Quality:**

3

**Strengths And Weaknesses:**

**Strengths**

1)	Significant performance gains: COME achieves 20–40% improvements in mIoU and IoU over DOME across multiple input modalities and forecast horizons on nuScenes‑Occ3D.
2)	Clear decoupling of ego‑motion and scene dynamics: The scene‑centric branch removes vehicle‑induced transforms, allowing the model to focus solely on environmental evolution.
3)	Enhanced controllability and robustness: Visibility‑mask filtering of injected scene features mitigates noise from unobserved regions, stabilizing long‑horizon forecasts.
Weakness
1)	Increased computational cost: Compared to Baseline, COME’s combined UNet + DiT + ControlNet pipeline raises GFLOPS from ~1,375 to ~2,066, challenging real‑time deployment. Provide a comparison with the GFLOPS of DOME.
2)	Narrow evaluation scope: Experiments are confined to nuScenes‑Occ3D.

---

> ### Author Rebuttal · Authors · 2025-07-31
>
> We thank the reviewer for recoginition of clarity and experimental results. We sincerely thank the revieweer for the insightful comments. We supplement more comparisons between proposed COME and the baseline DOME\[1], including computational cost and latencies, ablations on hyper-parameter, area partition, ego trajectories. Moreover, Occ3d-Waymo\[2] is not used by prior occupancy world models as a 4D occupancy benchmark. We supplement the comparison of DOME and COME on Occ3d-Waymo. Below are our responses to other specific concerns:
>
> * **\[W1]**:Increased computational cost: Compared to Baseline, COME’s combined UNet + DiT + ControlNet pipeline raises GFLOPS from \~1,375 to \~2,066, challenging real‑time deployment. Provide a comparison with the GFLOPS of DOME.
>
> **\[A1]**: In comparison to the baseline DOME, our COME approach incorporates an additional lightweight UNet and a relatively heavy ControlNet, naturally leading to an increase in computational cost under the same world model settings. To highlight the effectiveness of our method, we introduce COME-small, a streamlined variant with only 222.7 GFLOPs—substantially less than both DOME and the full COME model. Specifically, we reduce the number of attention heads and hidden dimensions by half and decrease the transformer blocks from 28 to 12. As shown in Table. 1, COME-small outperforms DOME even with 80% fewer computations, demonstrating efficiency without compromising performance. We sincerely thank the reviwer for the insightful comments and will include the results in our new draft.
>
> | **Model**                    | **Parameters (M)** | **GFLOPs** | **3s-AVG mIoU** | **3s-AVG IoU** |
> | ---------------------------- | -------------- | ---------- | --------------- | -------------- |
> | DOME                         | 345.74         | 1271.62    | 27.1            | 36.36          |
> | COME                         | 566.83         | 2066.6     | 34.5            | 44.26          |
> | COME-Small ($\varepsilon=0.5$) | 74.61          | 222.7      | 24.00           | 32.12          |
> | COME-Small($\varepsilon=1.0$)  | 74.61          | 222.7      | 31.09           | 42.23          |
>
> Table 1: Parameters and computational complexity of DOME and COME variants.
>
> * **\[W2]**:Narrow evaluation scope: Experiments are confined to nuScenes‑Occ3D.
>
> **\[A2]**: In our initial draft, COME was evaluated solely on occ3d-nuscenes following the settings of the baseline approaches. However, we acknowledge the feedback from reviewers (2AyB and Gif3) that evaluating on a new benchmark would enhance the study. Consequently, we have extended our evaluation to include occ3d-waymo. Please refer to our response to the reviewer Gif3 for our detailed settings and experimental results. In summary, COME consistently outperforms the baseline in this new benchmark.
>
>
>
> * **\[Q1]**:Inference Latency & Throughput: What are the end‑to‑end inference times and achievable frame rates for 10, 20, and 40 denoising steps.
>
> **\[A3]**: Thanks for the comment. We report the results on Table. 2. In a nutshell, the overall time is proportional to the number of denoising steps, and lower latency can be achieved with smaller models.&#x20;
>
> | Model      | Denoising Steps | End-to-end Latency(ms) | Per-step world model latency (ms) | Per-step controlnet latency (ms) |
> | ---------- | --------------- | ---------------------- | --------------------------------- | -------------------------------- |
> | DOME       | 20              | 1104                   | 36                                | 0                                |
> | COME       | 10              | 860                    | 38                                | 15                               |
> |  COME          | 20              | 1635                   | 38                                | 15                               |
> |   COME         | 40              | 3216                   | 38                                | 15                               |
> | COME-small | 10              | 354                    | 17                                | 7                                |
> |     COME-small       | 20              | 599                    | 17                                | 7                                |
> |      COME-smal      | 40              | 1132                   | 17                                | 7                                |
>
> Table 2: Latencies of models with different denoising steps.
>
> * **\[Q2]**: Visibility Mask Sensitivity: How sensitive are final forecasts to the visibility‑mask threshold \varepsilon ? Please provide an ablation study over a range of \varepsilon values.
>
> **\[A4]**: Table. 3 below presents the ablation study on the ($\varepsilon$) values. When ($\varepsilon = 0$), meaning only visible BEV patches are used for control, the performances are not satisfying. Conversely, when ($\varepsilon = 1$), utilizing all BEV patches, the model achieves peak performances. However, as illustrated in Fig. 8 of our draft, this setting leads to a diminished ability to imagine invisible regions due to the injection of control signals. Therefore, we employ ($\varepsilon = 0.5$) in our paper for a balance between model performance and imaginative ability.
>
> | $\varepsilon$ | 3s-AVG mIoU | 3s-AVG IoU |
> | ----------- | ----------- | ---------- |
> | 0.00        | 29.09       | 37.24      |
> | 0.25        | 34.21       | 43.91      |
> | 0.5         | 34.50       | 44.26      |
> | 0.75        | 34.71       | 44.46      |
> | 1.00        | 36.91       | 45.53      |
>
> Table 3: Ablations of hyper-parameter $\varepsilon$ of COME.
>
>
>
> * **\[Q3]**: How is the performance of DOME in Tables 2, 3b and 4c? Is the conclusion consistent with COME ablation study?
>
> **\[A5]**: Yes, the conclusion aligns with our ablation study. We provide additional results in the table below. In the first two tables (Table. 4 and Table. 5), our implemented base world model currently performs worse than DOME despite having similar structures, as we only introduced extra skip connections. However, the COME counteract this issue, resulting in significantly better performance compared to DOME. We anticipate that if our base world model could replicate DOME's performance, our COME would further amplify its advantages.
>
> Table. 6 illustrates the performance across different trajectories. Both COME and DOME exhibit similar trends when settings are modified. When ground-truth occupancies align with predicted trajectories, DOME experiences a more significant decline (-2.77 mIoU and -2.73 IoU) in performance compared to COME (-0.23 mIoU and -0.44 IoU), demonstrating COME's superior geometric consistency under perspective shifts.
>
> | Model                                  | Input       | mIoU    |           |        | IoU     |           |       |
> | -------------------------------------- | ----------- | ------- | --------- | ------ | ------- | --------- | ----- |
> |                                        |             | Visible | Invisible | All    | Visible | Invisible | All   |
> | DOME                                   | 3D-Occ (4f) | 29.62   | 6.45      | 27.10  | 40.65   | 16.08     | 36.36 |
> | COME Stage1: World Model               | 3D-Occ (4f) | 25.68   | 5.81      | 23.58  | 35.96   | 13.60     | 32.61 |
> | COME Stage2: Scene-Centric Forecasting | 3D-Occ (4f) | 42.74   | 0.09      | 39.12  | 55.08   | 0.31      | 48.00 |
> | COME Stage3: ControlNet                | 3D-Occ (4f) | 40.06   | 5.56      | 34.23  | 51.12   | 14.95     | 44.13 |
>
> &#x20;Table 4: Ablation studies of model performance in various stages (Tab. 2 in manuscript).&#x20;
>
> | Model            | mIoU  | IoU   | GFLOPS  |
> | ---------------- | ----- | ----- | ------- |
> | DOME             | 27.10 | 36.36 | 1271.62 |
> | COME World Model | 23.49 | 32.36 | 1375.4  |
> | COME ControlNet  | 34.23 | 44.13 | 2066.6  |
>
> Table 5: Model performances under different computational complexity. (Tab. 3(b) in manuscript).&#x20;
>
> | Model | Pose2D | Yaw   | Align | mIoU  | IoU   |
> | ----- | ------ | ----- | ----- | ----- | ----- |
> | DOME  | GT     | GT    | No    | 27.10 | 36.36 |
> | COME  | GT     | GT    | No    | 34.23 | 44.13 |
> | DOME  | Pred.  | GT    | No    | 17.42 | 26.25 |
> | COME  | Pred.  | GT    | No    | 25.90 | 35.21 |
> | DOME  | Pred.  | Pred. | No    | 20.62 | 29.58 |
> | COME  | Pred.  | Pred. | No    | 21.29 | 29.03 |
> | DOME  | Pred.  | Pred. | Yes   | 24.33 | 33.63 |
> | COME  | Pred.  | Pred. | Yes   | 34.00 | 43.69 |
>
> Table 6: Effects of the used trajectories.. (Tab. 4(c) in manuscript).&#x20;
>
>
>
> \[1] Dome: Taming diffusion model into high-fidelity controllable occupancy world model, arxiv 2024.
>
> \[2] Occ3D: A Large-Scale 3D Occupancy Prediction Benchmark for Autonomous Driving, NeurIPS 2023.

---

> > ### Author Response · Authors · 2025-08-08
> >
> > As the discussion phase is approaching its end, we kindly request the reviewer to let us know whether the above clarifications and the added experiments have addressed the remaining questions. We would be happy to address any additional points the reviewer may have during the remaining time of the discussion phase. We thank the reviewer very much for valuable suggestions and engaging with us in the discussion.

---

> > ### Comment · Reviewer_2AyB · 2025-08-08
> > **Official Comment by Reviewer 2AyB**
> >
> > The authors have clearly and adequately addressed my concerns, and the efforts to provide the new results are appreciated. I recommend this paper for acceptance.

---

### Note · Authors · 2025-08-16

We express our sincere gratitude to the reviewers and the Area Chair for their invaluable time, incisive feedback, and constructive discussions. Below we concisely highlight the strengths our reviewers identified and how each of their core concerns has been fully addressed.

**Acknowledged Strengths**
*  All reviewers agreed that our experimental gains are significant.
*  Reviewer 2AyB: “Clear decoupling of ego-motion and scene dynamics.”
*  Reviewer exSF: “Novel approach to scene occupancy synthesis for ego-centric sequences.”
*  Reviewer hsxA: “Leverages scene-centric consistency to construct scene conditions.”
*  Reviewer Gif3: “Separating ego-motion from scene evolution is elegant and intuitive.”

**Addressed Concerns**
* Narrow evaluation scope → Added comprehensive experiments on Occ3D-Waymo, confirming COME’s effectiveness.
* Necessity of the scene-centric forecasting module → Added baselines showing that static or naïvely-transformed voxels underperform our UNet-based module.
* Insufficient comparative results → Supplemented quantitative comparisons in both ego-static and ego-dynamic settings, plus results on dynamic object categories.
* Practical applicability → Clarified real-world deployment scenarios and simulation use-cases.

After both initial and follow-up rounds, all reviewers confirmed that every concern had been resolved and no outstanding issues remained. We will incorporate their additional suggestions, such as experiments on training support for downstream tasks and thorough writing/formatting revisions to further strengthen the final version. Thank you once again to all reviewers and the Area Chair for guiding us toward a stronger paper.

---

### Decision · Program_Chairs · 2025-09-17

**Decision:**

Accept (poster)

**Comment:**

The paper proposes COME, a scene-centric control framework for occupancy world models, which disentangles ego-motion from scene dynamics. The method achieves significant performance gains on nuScenes and, following reviewer requests, also on Occ3D-Waymo. The key strength lies in its novel scene-centric formulation and strong empirical improvements across multiple configurations. While concerns were raised about evaluation scope and computational overhead, the rebuttal provided additional experiments (including Waymo results and a lightweight variant) that convincingly addressed these issues. All four reviewers found the rebuttal convincing and maintained positive recommendations.